# A multi-task convolutional deep neural network for variant calling in single molecule sequencing

Ruibang Luo [1,2], Fritz J. Sedlazeck[3], Tak-Wah Lam[1] & Michael C. Schatz[2]

The accurate identification of DNA sequence variants is an important, but challenging task in genomics. It is particularly difficult for single molecule sequencing, which has a per-nucleotide error rate of ~5–15%. Meeting this demand, we developed Clairvoyante, a multi-task five-layer convolutional neural network model for predicting variant type (SNP or indel), zygosity, alternative allele and indel length from aligned reads. For the well-characterized NA12878 human sample, Clairvoyante achieves 99.67, 95.78, 90.53% F1-score on 1KP common variants, and 98.65, 92.57, 87.26% F1-score for whole-genome analysis, using Illumina, PacBio, and Oxford Nanopore data, respectively. Training on a second human sample shows Clairvoyante is sample agnostic and finds variants in less than 2 h on a standard server. Furthermore, we present 3,135 variants that are missed using Illumina but supported independently by both PacBio and Oxford Nanopore reads. Clairvoyante is available open-source (https://github.com/aquaskyline/Clairvoyante), with modules to train, utilize and visualize the model.

---

[1] Department of Computer Science, The University of Hong Kong, Hong Kong 999077, China. [2] Department of Computer Science, Johns Hopkins University, Baltimore 21218 MD, USA. [3] Human Genome Sequencing Center, Baylor College of Medicine, Houston 77030 TX, USA. Correspondence and requests for materials should be addressed to R.L. (email: rbluo@cs.hku.hk)

A fundamental problem in genomics is to find the nucleotide differences in an individual genome relative to a reference sequence, i.e., variant calling. It is essential to accurately and efficiently call variants so that the genomic variants that underlie phenotypic differences and disease can be correctly detected[1]. Previous works have intensively studied the different data characteristics that might contribute to higher variant calling performance, including the properties of the sequencing instrument[2], the quality of the preceding sequence aligners[3], and the alignability of the genome reference[4]. Today, these characteristics are carefully considered by state-of-the-art variant calling pipelines to optimize performance[5,6]. However, most of these analyses were done for short-read sequencing, especially the Illumina technology, and require further study for other sequencing platforms.

Single-molecule sequencing (SMS) technologies are emerging in recent years for a variety of important applications[7]. These technologies generate sequencing reads, which are two orders of magnitude longer than standard short-read Illumina sequencing (10–100 kbp instead of ~100 bp), but they also contain 5–15% sequencing errors rather than ~1% for Illumina. The two major SMS companies, Pacific Biosciences (PacBio) and Oxford Nanopore Technology (ONT) have greatly improved the performance of certain genomic applications, especially genome assembly and structural variant detection[7]. However, single nucleotide and small indel variant calling with SMS remain challenging because the traditional variant caller algorithms fail to handle such a high-sequencing error rate, especially the one enriched for indel errors.

Artificial neural networks (ANNs) are becoming increasingly prominent for a variety of classification and analysis tasks due to their advances in speed and applicability in many fields. One of the most important applications of ANNs has been image classification, with many successes, including MNIST[8] or GoogLeNet[9]. The recent DeepVariant[10] package repurposed the inception convolutional neural network for DNA variant detection by applying it to analyzing images of aligned reads around candidate variants. At each candidate site, the network computes the probabilities of three possible zygosities (homozygous reference, heterozygous reference, and homozygous alternative), allowing accurate determination of the presence or absence of a candidate variant. Then, Deep-Variant uses a post-processing step to restore the other variant information, including the exact alternative allele and variant type. As the authors pointed out originally in their paper, it might be suboptimal to use an image classifier for variant calling, as valuable information that could contribute to higher accuracy is lost during the image transformation. In the latest version of DeepVariant, the code is built on top of the Tensorflow machine-learning framework, allowing users to change the image input into any other formats by rewriting a small part of the code. However, whether it is reasonable or not to use a network (namely inception-v3) specifically designed for image-related tasks to call variants remains unclear.

In this study, we present Clairvoyante, a multitask convolutional deep neural network specifically designed for variant calling with SMS reads. We explored different ways to enhance Clairvoyante's power to extract valuable genomic features from the frequent background errors present in SMS. Experiments calling variants in multiple human genomes both at common variant sites and genome-wide show that Clairvoyante is on par with GATK UnifiedGenotyper on Illumina data, and substantially outperforms Nanopolish and DeepVariant on PacBio and ONT data on accuracy and speed.

**Table 1 Time per epoch of different models of GPU and CPU in model training**

| Equipment | Seconds per epoch per 11 M samples |
|---|---|
| GTX 1080 Ti | 170 |
| GTX 980 | 250 |
| GTX Titan | 520 |
| Tesla K40 w/top power setting | 580 |
| Tesla K40 | 620 |
| Tesla K80 (one socket) | 700 |
| GTX 680 | 780 |
| Intel Xeon E5-2680 v4 28-core | 2900 |

## Results

**Overview**. In this section, we first benchmarked Clairvoyante on Illumina, PacBio, and ONT data at the common variant sites from 1000 Genomes Project phase 3[11] with a minor allele frequency ≥5%. Then, we evaluated Clairvoyante's performance to call variants genome-wide. In addition, we explored and benchmarked other state-of-the-art variant callers that were not primarily designed for SMS variant calling. Finally, we carried out unit tests and answered the following questions in Supplementary Note, Unit tests, including (1) What are the characteristics of false positives and false negatives? (2) Can lower learning rate and longer training provide better performance? (3) Can a model train on truth variants from multiple samples provide better performance? (4) Can a higher input data quality improve the variant calling performance? (5) Is the current network design sufficient in terms of learning capacity?

**Training runtime performance**. We recommend using graphics processing unit (GPU) acceleration for model training and central processing unit (CPU)-only for variant calling. Table 1 shows the performance of different GPU and CPU models in training. Using a high-performance desktop GPU model GTX 1080 Ti, 170 s are needed per epoch, which leads to about 5 h to finish training a model with the fast training mode. However, for variant calling, the speedup by GPU is insignificant because CPU workloads such as VCF file formatting and I/O operations dominate. Variant calling at 8.5 M common variant sites takes about 40 min using 28 CPU cores. Variant calling genome-wide varies between 30 min and a few hours subject to which sequencing technology and alternative allele frequency cutoff were used.

**Call variants at common variant sites**. Clairvoyante was designed targeting SMS; nevertheless, the method is generally applicable for short-read data as well. We benchmarked Clairvoyante on three sequencing technologies: Illumina, PacBio, and ONT using both the fast and the nonstop training mode. In the nonstop training mode, we started training the model from 0 to 999-epoch at learning rate $1e^{-3}$, then to 1499-epoch at $1e^{-4}$, and finally to 1999-epoch at $1e^{-5}$. We then benchmarked the model generated by the fast mode, and all three models stopped at different learning rates in the nonstop mode. We also benchmarked variant calling on one sample (e.g., HG001) using a model trained on another sample (e.g., HG002). Further, we ran GATK UnifiedGenotyper[6] and GATK HaplotypeCaller[6] for comparison. Noteworthy, GATK UnifiedGenotyper was superseded by GATK HaplotypeCaller; thus, for Illumina data, we should refer to the results of HaplotypeCaller as the true performance of GATK. However, our benchmarks show that

UnifiedGenotyper performed better than HaplotypeCaller on the PacBio and ONT data; thus, we also benchmarked UnifiedGenotyper for all three technologies for users to make parallel comparisons. We also attempted to benchmark other tools for SMS reads, including PacBio GenomicConsensus v5.1[12], and Nanopolish v0.9.0[13], but we only completed the benchmark with Nanopolish. The reason why GenomicConsensus failed, and the commands used for generating the results in this section are presented in Supplementary Note, Call Variants at Common Variant Sites Commands. We also benchmarked DeepVariant[10] and LoFreq[14]. Although the two tools were primarily designed to work with Illumina short reads, we managed to run them with ONT long reads. The reason why the two tools failed on PacBio is elaborated in the Section "Benchmarks of other state-of-the-art variant callers".

The benchmarks at the common variant sites from 1000 Genomes Project[11] phase 3 with global minor allele frequency ≥5% (8,511,819 sites for GRCh37, 8,493,490 sites for GRCh38, hereafter referred to as "1KGp3") demonstrate the expected performance of Clairvoyante on a typical precision medicine application that only tens to hundreds of known clinically relevant or actionable variants are being genotyped. This is becoming increasingly important in recent days as SMS is becoming more widely used for clinical diagnosis of structural variations, but at the same time, doctors and researchers also want to know if there exist any actionable or incidental small variants without additional short-read sequencing[15]. So first, we have evaluated Clairvoyante's performance on common variant sites before extending the evaluation genome-wide. The latter is described in the Section "Genome-wide variant identification".

We used the submodule *vcfeval* in RTG Tools[16] version 3.7 to benchmark our results and generate five metrics, including false-positive rate (FPR), false-negative rate (FNR), Precision, Recall, and F1-score. From the number of true positives (*TP*), false positives (*FP*), true negatives (*TN*), and false negatives (*FN*), we compute the five metrics as $\text{FPR} = FP \div (FP + TN)$, $\text{FNR} = FN \div (FN + TP)$, $\text{Precision} = TP \div (TP + FP)$, $\text{Recall} = TP \div (TP + FN)$, and $\text{F1-score} = 2TP/(2TP + FN + FP)$. As FNR can be calculated as 1 minus the Recall, we only used FNR in this section as readers can deduce the corresponding Recall easily. *TP* are defined as variants existing in both the 1KGp3 and GIAB dataset that identified as a variant by Clairvoyante with no

discrepancy in terms of allele, type, zygosity, and indel length if applicable. *TN* are defined as variants existing in 1KGp3 but not in the GIAB dataset that identified as a nonvariant by Clairvoyante. *FP* are defined as (1) sites supposed to be *TN* but identified by Clairvoyante as a variant, or (2) variants existing in the GIAB dataset that also identified as a variant by Clairvoyante, but with discrepant variant type, alternative allele, or zygosity. *FN* are defined as the variants existing in the GIAB dataset but identified as a nonvariant by Clairvoyante. F1-score is the harmonic mean of the precision and recall. RTG *vcfeval* also provides the best variant quality cutoff for each dataset, filtering the variants under which they can maximize the F1-score. To the best of our knowledge, RTG *vcfeval* was also used by the GIAB project itself. *vcfeval* cannot deal with Indel variant calls without an exact allele. However, in our study, Clairvoyante was set to provide the exact allele only for Indels ≤ 4 bp. Thus, for Clairvoyante, all Indels > 4 bp were removed from both the baseline and the variant calls before benchmarking. The commands used for benchmarking are presented in Supplementary Note, Benchmarking Commands.

Table 2 shows the performance of Clairvoyante on Illumina data. The best accuracy is achieved by calling variants in HG001 using the model trained on HG001 at 999-epoch, with 0.25% FPR, 0.41% FNR, 99.75% precision, and 99.67% F1-score. A major concern of using deep learning or any statistical learning technique for variant calling is the potential for overfitting to the training samples. Our results show that Clairvoyante is not affected by overfitting, and we validated the versatility of the trained models by calling variants in a genome using a model trained on a second sample. Interestingly, the F1-score of calling variants in HG002 using a model trained on HG001 (for convenience, hereafter denoted as HG002 > HG001) is 0.09% higher (99.61% against 99.52%) than HG002 > HG002 and similar to HG001 > HG001. As we know that the truth variants in HG001 were verified and rectified by more orthogonal genotyping methods than HG002[17], we believe that it is the higher quality of truth variants in HG001 than HG002 that gave the model trained on HG001 a higher performance. The gap of FNR between Clairvoyante and GATK UnifiedGenotyper on HG001 is 0.68% (3.11% against 2.43%) but enlarged to 3.27% (5.80% against 2.52%) on HG002. This again corroborated the importance of high-quality truth variants for Clairvoyante to achieve superior performance.

### Table 2 Performance of Clairvoyante on Illumina data at common variant sites in 1KGp3

| Seq. tech. | Model trained on | Trained epochs | Ending learning rate and lambda | Call variants in | Best variant quality cutoff | Overall FPR (%) | Overall FNR (%) | Overall precision (%) | Overall F1 score (%) | SNP FPR (%) | SNP FNR (%) | SNP precision (%) | SNP F1 score (%) | Indel FPR (%) | Indel FNR (%) | Indel precision (%) | Indel F1 score (%) |
|---|---|---|---|---|---|---|---|---|---|---|---|---|---|---|---|---|---|
| Illumina | HG001 | 67[a] | 1.E−05 | HG001 | 67 | 0.28 | 0.45 | 99.72 | 99.64 | 0.07 | 0.10 | 99.93 | 99.91 | 1.93 | 3.38 | 98.00 | 97.31 |
| | | 999 | 1.E−03 | | 119 | 0.25 | 0.41 | 99.75 | 99.67 | 0.08 | 0.08 | 99.92 | 99.92 | 1.64 | 3.13 | 98.30 | 97.58 |
| | | 1499 | 1.E−04 | | 128 | 0.28 | 0.41 | 99.72 | 99.66 | 0.08 | 0.08 | 99.92 | 99.92 | 1.87 | 3.11 | 98.07 | 97.48 |
| | | 1999 | 1.E−05 | | 147 | 0.29 | 0.42 | 99.71 | 99.64 | 0.08 | 0.09 | 99.92 | 99.92 | 1.95 | 3.24 | 97.98 | 97.37 |
| | HG001 | 67[a] | 1.E−05 | HG002 | 58 | 0.32 | 0.51 | 99.68 | 99.59 | 0.11 | 0.15 | 99.89 | 99.87 | 2.11 | 3.58 | 97.82 | 97.12 |
| | | 999 | 1.E−03 | | 107 | 0.30 | 0.49 | 99.70 | 99.61 | 0.11 | 0.14 | 99.89 | 99.87 | 1.94 | 3.47 | 97.99 | 97.26 |
| | | 1499 | 1.E−04 | | 151 | 0.34 | 0.54 | 99.66 | 99.56 | 0.11 | 0.16 | 99.89 | 99.86 | 2.26 | 3.80 | 97.69 | 96.92 |
| | | 1999 | 1.E−05 | | 147 | 0.37 | 0.54 | 99.63 | 99.55 | 0.12 | 0.15 | 99.88 | 99.86 | 2.46 | 3.89 | 97.45 | 96.77 |
| | HG002 | 66[a] | 1.E−05 | HG001 | 53 | 0.31 | 0.80 | 99.69 | 99.44 | 0.08 | 0.14 | 99.92 | 99.89 | 2.18 | 6.26 | 97.68 | 95.67 |
| | | 999 | 1.E−03 | | 96 | 0.28 | 0.76 | 99.72 | 99.48 | 0.07 | 0.13 | 99.93 | 99.90 | 2.00 | 6.00 | 97.87 | 95.90 |
| | | 1499 | 1.E−04 | | 134 | 0.33 | 0.81 | 99.67 | 99.43 | 0.08 | 0.15 | 99.92 | 99.88 | 2.37 | 6.34 | 97.48 | 95.53 |
| | | 1999 | 1.E−05 | | 148 | 0.35 | 0.83 | 99.65 | 99.41 | 0.08 | 0.15 | 99.92 | 99.89 | 2.50 | 6.50 | 97.34 | 95.38 |
| | HG002 | 66[a] | 1.E−05 | HG002 | 54 | 0.28 | 0.76 | 99.72 | 99.48 | 0.07 | 0.13 | 99.93 | 99.90 | 2.01 | 6.17 | 97.86 | 95.80 |
| | | 999 | 1.E−03 | | 99 | 0.24 | 0.72 | 99.76 | 99.52 | 0.06 | 0.13 | 99.94 | 99.90 | 1.76 | 5.80 | 98.14 | 96.13 |
| | | 1499 | 1.E−04 | | 124 | 0.27 | 0.72 | 99.73 | 99.50 | 0.07 | 0.12 | 99.93 | 99.90 | 1.96 | 5.87 | 97.92 | 95.99 |
| | | 1999 | 1.E−05 | | 132 | 0.28 | 0.73 | 99.72 | 99.50 | 0.07 | 0.12 | 99.93 | 99.90 | 2.03 | 5.94 | 97.84 | 95.91 |
| DeepVariant | | | | | 3 | 0.04 | 0.06 | 99.96 | 99.95 | 0.01 | 0.03 | 99.99 | 99.98 | 0.27 | 0.28 | 99.72 | 99.72 |
| LoFreq | | | | | | Benchmarked SNP only | | | | 10.59 | 0.51 | 85.02 | 91.69 | – | – | – | – |
| GATK UnifiedGenotyper, HG001 | | | | | 3 | 0.19 | 0.35 | 99.81 | 99.73 | 0.10 | 0.07 | 99.90 | 99.91 | 0.80 | 2.43 | 99.14 | 98.35 |
| GATK HaplotypeCaller, HG001 | | | | | 4 | 0.07 | 0.11 | 99.93 | 99.91 | 0.013 | 0.03 | 99.99 | 99.98 | 0.50 | 0.66 | 99.47 | 99.41 |
| GATK UnifiedGenotyper, HG002 | | | | | 3 | 0.20 | 0.37 | 99.80 | 99.71 | 0.12 | 0.09 | 99.87 | 99.89 | 0.73 | 2.52 | 99.20 | 98.33 |
| GATK HaplotypeCaller, HG002 | | | | | 5 | 0.07 | 0.10 | 99.93 | 99.92 | 0.023 | 0.05 | 99.98 | 99.96 | 0.38 | 0.50 | 99.60 | 99.55 |

[a]Fast training mode

**Table 3 Performance of Clairvoyante on PacBio data at common variant sites in 1KGp3**

| Seq. tech. | Model trained on | Trained epochs | Ending learning rate and lambda | Call variants in | Best variant quality cutoff | Overall FPR (%) | Overall FNR (%) | Overall precision (%) | Overall F1 score (%) | SNP FPR (%) | SNP FNR (%) | SNP precision (%) | SNP F1 score (%) | Indel FPR (%) | Indel FNR (%) | Indel precision (%) | Indel F1 score (%) |
|---|---|---|---|---|---|---|---|---|---|---|---|---|---|---|---|---|---|
| PacBio | HG001 | 50[a] | 1.E−05 | HG001 | 69 | 1.51 | 7.41 | 98.38 | 95.39 | 0.32 | 1.43 | 99.68 | 99.12 | 10.94 | 60.59 | 76.24 | 51.96 |
| | | 999 | 1.E−03 | | 94 | 1.39 | 7.07 | 98.51 | 95.64 | 0.26 | 1.29 | 99.74 | 99.22 | 10.39 | 58.41 | 78.21 | 54.31 |
| | | 1499 | 1.E−04 | | 89 | 2.17 | 6.06 | 97.70 | 95.78 | 0.25 | 1.18 | 99.75 | 99.28 | 16.44 | 49.38 | 72.02 | 59.45 |
| | | 1999 | 1.E−05 | | 85 | 2.43 | 5.81 | 97.43 | 95.78 | 0.26 | 1.18 | 99.74 | 99.28 | 18.20 | 46.98 | 70.44 | 60.50 |
| | HG001 | 50[a] | 1.E−05 | HG002 | 75 | 1.78 | 7.48 | 98.07 | 95.22 | 0.71 | 1.47 | 99.28 | 98.91 | 10.29 | 60.05 | 77.70 | 52.77 |
| | | 999 | 1.E−03 | | 96 | 1.98 | 7.31 | 98.01 | 95.21 | 0.75 | 1.45 | 99.23 | 98.89 | 11.54 | 58.54 | 76.08 | 53.67 |
| | | 1499 | 1.E−04 | | 114 | 2.07 | 7.77 | 97.76 | 94.91 | 0.76 | 1.45 | 99.23 | 98.89 | 12.21 | 63.04 | 72.67 | 49.00 |
| | | 1999 | 1.E−05 | | 123 | 1.97 | 7.94 | 97.86 | 94.87 | 0.75 | 1.44 | 99.24 | 98.90 | 11.50 | 64.74 | 73.09 | 47.57 |
| | HG002 | 72[a] | 1.E−05 | HG001 | 56 | 1.63 | 9.22 | 98.20 | 94.35 | 0.49 | 2.55 | 99.49 | 98.46 | 10.72 | 68.46 | 72.43 | 43.94 |
| | | 999 | 1.E−03 | | 99 | 1.69 | 8.47 | 98.16 | 94.73 | 0.57 | 1.84 | 99.42 | 98.79 | 10.62 | 67.39 | 73.31 | 45.14 |
| | | 1499 | 1.E−04 | | 116 | 2.43 | 8.25 | 97.36 | 94.47 | 0.80 | 1.91 | 99.19 | 98.64 | 14.89 | 64.53 | 66.97 | 46.37 |
| | | 1999 | 1.E−05 | | 127 | 2.34 | 8.57 | 97.45 | 94.34 | 0.89 | 2.04 | 99.09 | 98.52 | 13.56 | 66.58 | 68.06 | 44.83 |
| | HG002 | 72[a] | 1.E−05 | HG002 | 55 | 1.88 | 7.08 | 97.98 | 95.38 | 0.55 | 1.25 | 99.44 | 99.10 | 12.15 | 58.10 | 75.20 | 53.82 |
| | | 999 | 1.E−03 | | 88 | 1.86 | 6.59 | 98.01 | 95.66 | 0.49 | 1.15 | 99.51 | 99.18 | 12.45 | 54.11 | 76.34 | 57.32 |
| | | 1499 | 1.E−04 | | 101 | 2.02 | 5.81 | 97.85 | 95.99 | 0.42 | 1.02 | 99.57 | 99.28 | 14.10 | 47.73 | 76.11 | 61.98 |
| | | 1999 | 1.E−05 | | 101 | 2.05 | 5.70 | 97.83 | 96.03 | 0.41 | 0.99 | 99.59 | 99.30 | 14.40 | 46.87 | 75.96 | 62.52 |
| GATK UnifiedGenotyper, HG001 | | | | | 1 | 0.83 | 99.92 | 8.19 | 0.15 | 0.94 | 99.91 | 8.19 | 0.17 | – | – | – | – |
| GATK HaplotypeCaller, HG001 | | | | | 1 | 0.08 | 97.91 | 52.26 | 4.02 | 0.06 | 97.65 | 61.16 | 4.53 | 0.67 | 99.98 | 0.39 | 0.04 |
| GATK UnifiedGenotyper, HG002 | | | | | 1 | 0.75 | 99.91 | 9.91 | 0.17 | 0.85 | 99.90 | 9.91 | 0.19 | – | – | – | – |
| GATK HaplotypeCaller, HG002 | | | | | 1 | 0.69 | 98.74 | 63.81 | 2.47 | 0.79 | 98.58 | 63.88 | 2.78 | 0.02 | 100.00 | 15.66 | 0.007 |

[a]Fast training mode

**Table 4 Performance of Clairvoyante on ONT data at common variant sites in 1KGp3**

| Seq. Tech. | Model trained on | Trained epochs | Ending learning rate and lambda | Call variants in | Best variant quality cutoff | Overall FPR (%) | Overall FNR (%) | Overall precision | Overall F1 score (%) | SNP FPR (%) | SNP FNR (%) | SNP precision (%) | SNP F1 score (%) | Indel FPR (%) | Indel FNR (%) | Indel precision (%) | Indel F1 score (%) |
|---|---|---|---|---|---|---|---|---|---|---|---|---|---|---|---|---|---|
| ONT | HG001 (except for chr1) | 110[a] | 1.E−05 | HG001 (chr1) | 47 | 3.40 | 17.09 | 95.93 | 88.94 | 3.33 | 9.29 | 96.34 | 93.44 | 3.99 | 86.42 | 76.59 | 23.07 |
| | | 999 | 1.E−03 | | 53 | 4.05 | 16.31 | 95.20 | 89.07 | 3.79 | 8.95 | 95.85 | 93.39 | 6.28 | 81.70 | 73.19 | 29.28 |
| | | 1499 | 1.E−04 | | 70 | 2.95 | 14.99 | 96.55 | 90.41 | 2.55 | 7.76 | 97.24 | 94.68 | 6.32 | 79.28 | 75.46 | 32.51 |
| | | 1999 | 1.E−05 | | 74 | 2.96 | 14.78 | 96.55 | 90.53 | 2.52 | 7.59 | 97.28 | 94.78 | 6.68 | 78.64 | 74.90 | 33.24 |
| Nanopolish, HG001 | | | | | 20 | 0.04 | 21.57 | 97.51 | 86.93 | 0.03 | 15.66 | 98.11 | 90.70 | 0.12 | 63.46 | 88.57 | 51.73 |
| DeepVariant (chr1) | | | | | 3 | 0.22 | 23.82 | 96.26 | 85.05 | 0.21 | 15.30 | 96.78 | 90.33 | 0.30 | 89.11 | 72.98 | 18.95 |
| LoFreq | | | | | – | Benchmarking SNP only | | | | 2.79 | 46.99 | 90.69 | 66.91 | – | – | – | – |
| GATK UnifiedGenotyper, HG001 | | | | | 1 | 3.66 | 84.44 | 80.07 | 26.05 | 4.12 | 82.47 | 80.07 | 28.77 | – | – | – | – |
| GATK HaplotypeCaller, HG001 | | | | | 1 | 0.41 | 98.65 | 76.04 | 2.65 | 0.45 | 98.48 | 76.73 | 2.97 | 0.14 | 99.98 | 9.59 | 0.03 |
| Nanopolish, afcut0.2, depthcut4, and chr19 | | | | | 20 | 0.15 | 34.13 | 90.00 | 76.07 | 0.08 | 27.23 | 94.56 | 82.25 | 0.73 | 83.06 | 36.45 | 23.13 |
| Nanopolish, 1kgp3, and chr19 | | | | | 20 | 0.08 | 22.71 | 95.28 | 85.35 | 0.05 | 16.49 | 96.88 | 89.70 | 0.30 | 66.77 | 73.64 | 45.79 |

[a]Fast training mode

Table 3 shows the performance of Clairvoyante on PacBio data. The best performance is achieved by calling variants in HG001 using the model trained on HG001 at 1499-epoch, with 2.17% FPR, 6.06% FNR, 97.70% precision, and 95.78% F1-score. As previously reported, DeepVariant[10] has benchmarked the same dataset in their study and reported 97.25% precision, 88.51% recall (11.49% FNR), and 92.67% F1-score. We noticed that our benchmark differs from DeepVariant because we have removed Indels >4 bp (e.g., 52,665 sites for GRCh38 and 52,709 for GRCh37 in HG001) from both the baseline and variant calls. If we assume that DeepVariant can identify all the 91k Indels >4 bp correctly, it's recall will increase to 90.73% (9.27% FNR), which is still 3.21% lower than Clairvoyante. Clairvoyante's SNP calling F1-score on PacBio data topped at 99.28% for HG001 and 99.30% for HG002, making Clairvoyante suitable for genotyping SNPs sensitively and accurately at known clinically relevant or actionable variants using PacBio reads in precision medicine application.

Table 4 shows the performance of Clairvoyante on ONT data. As there are no publicly available deep coverage ONT datasets for HG002, we benchmarked variant calls in the chromosome 1 of HG001 using models trained on all chromosomes of HG001 except for the chromosome 1. The best F1-score, 90.53%, is achieved at 1999-epoch, with FPR 2.96%, FNR 14.78%, and precision 96.55%. For comparison, we have applied Nanopolish to the whole genome of HG001. Using 24 CPU cores and a peak of 160-GB memory, it finished in 4.5 days and achieved 0.04, 21.57, 97.51%, and 86.93% on FPR, FNR, precision, and F1-score, respectively.

**Genome-wide variant identification**. Beyond benchmarking variants at sites known to be variable in a sample, in this section, we benchmarked Clairvoyante's performance on calling variants genome-wide. Calling variants genome-wide is challenging because it tests not only how good Clairvoyante can derive the correct variant type, zygosity, and alternative allele of a variant when evidence is marginal, but also in reverse, how good Clairvoyante can filter/suppress a nonvariant even in the presence of sequencing errors or other artificial signals. Instead of naively evaluating all 3 billion sites of the whole genome with Clairvoyante, we tested the performance at different alternative allele cutoffs for all three sequencing technologies. As expected, a higher allele cutoff speeds up variant calling by producing fewer candidates to be tested by Clairvoyante but worsens recall, especially for noisy data like PacBio and ONT. Our experiments provide a reference point on how to choose a cutoff for each sequencing technology to achieve a good balance between recall and running speed. We used Precision, Recall, and F1-score metrics but we did not use FPR (calculated as $FP \div (FP + TN)$) in this section because $FP$ becomes negligibly small compared with $TN$, which is around 3 billion in human genome-wide variant identification. All models were trained for 1000 epochs with a learning rate at $1e^{-3}$. All the experiments were performed on two

Intel Xeon E5-2680 v4 using all 28 cores. The commands used for generating the results in this section are presented in Supplementary Note, Call Variants Genome-wide Commands.

The results are shown in Supplementary Table 1. As expected, with a higher alternative allele frequency threshold (0.2), the precision was higher, while the recall and time consumption was reduced in all experiments. For Illumina data, the best F1-score (with 0.2 allele frequency) for Clairvoyante was 98.65% for HG001 and 98.61% for HG002. The runtime varied between half and an hour (40 min for the best F1-score). As expected, GATK HaplotypeCaller topped the performance on Illumina data—it achieved F1-scores of 99.76% for HG001 and 99.70% for HG002; both ran for about 8 h. GATK UnifiedGenotyper ran as fast as Clairvoyante on Illumina data and achieved F1-scores of 99.43% for HG001 and 99.08% for HG002. Inspecting the false-positive and false-negative variant calls for Clairvoyante, we found that about 0.19% in FP, and 0.15% in FN was because of scenarios of two alternative alleles.

We realized, on Illumina data, that Clairvoyante is not performing on par with the state-of-the-art GATK Haplotype-Caller, which was intensively optimized for Illumina data. However, as Clairvoyante uses an entirely different algorithm than GATK, Clairvoyante's architecture could be used as an orthogonal method, emulating how geneticists manually validate a variant using a genome browser, for filtering or validating GATK's results to increase GATK's accuracy further. We implemented this in a method called Skyhawk. It repurposed Clairvoyante's neural network to work on the GATK's variants, give them another quality score in addition to the existing one by GATK, and give suggestion on disagreed answers. More details are available in Skyhawk's preprint[18]. With the success of developing Skyhawk, we expect to see in the future, that more applications would be developed upon Clairvoyante's network architecture.

For the PacBio data, the best F1-scores were also achieved at 0.2 allele frequency cutoff. The best F1-score is 92.57% for HG001 and 93.05% for HG002 running Clairvoyante for ~3.5 h. In contrast, as reported in their paper[10], DeepVariant has achieved 35.79% F1-score (22.14% precision, 93.36% recall) on HG001 with PacBio data. The runtime for Clairvoyante at 0.25 frequency cutoff is about 2 h, which is about half the time consumption at 0.2 frequency cutoff, and about 1/5 the time consumption at 0.1 frequency cutoff. For ONT data (rel3), the best F1-score 77.89% was achieved at 0.1 frequency cutoff. However, the F1-score at 0.25 frequency cutoff is just slightly lower (76.95%), but ran about five times faster, from 13 h to less than 3 h. Thus, we suggest using 0.25 as the frequency cutoff. The runtime is on average about 1.5 times longer than PacBio, due to the higher level of noise in data. Using the new rel5 ONT data with better base-calling quality, the best F1-score has increased from 87.26% (9.37% higher than rel3). The recall of SNP and the precision of Indel were the most substantially increased.

For readers to compare the whole-genome benchmarks to those at the common variant sites more efficiently, we summarized the best precision, recall, and F1-score of both types of benchmarks in Supplementary Table 2.

**Benchmarks of other state-of-the-art variant callers**. DeepVariant is the first deep neural network-based variant caller[10]. After the first preprint of Clairvoyante was available, Google released a new version of DeepVariant (v0.6.1). On Illumina data, the new version was reported to be outperforming the previous versions. We benchmarked the new version to see how it performs on Illumina data and especially on SMS data. We used DeepVariant version 0.6.1 for benchmarking following the guide "Improve

DeepVariant for BGISEQ germline variant calling" written by DeepVariant coauthor Pi-Chuan Chang available at the link https://goo.gl/tg4FWG with specific guidelines on how to run DeepVariant, including (1) model training using transfer-learning and multiple depths, and (2) variant calling.

On Illumina data, DeepVariant performed extraordinarily (Supplementary Table 3) and matched with the figures previously reported. Following the guide, we applied transfer learning using both the truth variants and reference calls in chromosome 1 upon the trained model named "DeepVariant-inception_v3-0.6.0 + cl-191676894.data-wgs_standard/model.ckpt" that was delivered together with the software binaries. Using a nVidia GTX1080 Ti GPU, we kept running the model training process for 24 h and picked the model with the best F1-score (using chromosome 22 for validation purpose), which was achieved at about 65 min after the training had started. The variant calling step comprises three steps: (1) create calling candidates, (2) variant calling, and (3) post processing. Using 24 CPU cores, step 1 ran for 392 min and generated 42 GB of data. The second step utilized GPU and took 166 min. Step 3 ran for only 25 min and occupied significantly more memory (15 GB) than the previous two steps. For the HG001 sample, the precision rate is 0.9995, and the recall rate is 0.9991, both extraordinary and exceeding all other available variant callers, including Clairvoyante on Illumina datasets.

DeepVariant requires base quality, and thus failed on the PacBio dataset, in which base quality is not provided. On ONT data (rel5), DeepVariant performed much better than the traditional variant callers that were not designed for long reads, but it performed worse than Clairvoyante (Supplementary Table 3). We also found that DeepVariant's computational resource consumption on long reads is prohibitively high and we were only able to call variants in few chromosomes. The details are as follows. Using transfer learning, we trained two models for ONT data on chromosomes 1 and 21, respectively, and we called variants in chromosomes 1 and 22 against the different models. In total, we have benchmarked three settings, (1) call variants in chromosome 1 against the chromosome 21 model, (2) call variants in chromosome 22 against the chromosome 21 model, and (3) call variants in chromosome 22 against the chromosome 1 model. Training the models required about 1.5 days until the validation showed a decreasing F1-score with further training. Using 24 CPU cores, the first step of variant calling generated 337-GB candidate variant data in 1,683 min for chromosome 1 and generated 53-GB data in 319 min for chromosome 21. The second step of variant calling took 1,171 and 213 min to finish for chromosomes 1 and 22, respectively. The last step took 160 min and was very memory intensive, requiring 74 GB of RAM for chromosome 1. In terms of the F1-score, DeepVariant has achieved 83.05% in chromosome 1, and 77.89% in chromosome 22, against the model trained on chromosome 21. We verified that more samples for model training do not lead to better variant calling performance—using the model trained on chromosome 1, the F1-score dropped slightly to 77.09% for variants in chromosome 22. Using the computational resource consumption on chromosome 1, we estimate that the current version of DeepVariant would require 4-TB storage and about 1 month for whole-genome variant calling of a genome sequenced with long reads.

We further benchmarked three additional variant callers[19], including Vardict[20] (v20180724), LoFreq[14] (v2.1.3.1), and Free-Bayes[21] (v1.1.0-60-gc15b070) (Supplementary Table 3). The performance of Vardict on Illumina data matches the previous study[19]. Vardict requires base quality, and thus failed on the PacBio dataset, in which base quality is not provided. Vardict identified only 62,590 variants in the ONT dataset; among them, only 231 variants are true positives. The results match with

Vardict's paper that was tested on the Illumina data but not yet ready for Single Molecule Sequencing long reads. The performance of LoFreq on Illumina data matches the previous study[19] calling SNP only. To enable Indel calling in LoFreq, BAQ (Base Alignment Quality)[22] needs to be calculated in advance. However, the BAQ calculation works only for Illumina reads; thus, for LoFreq, we only benchmarked its performance in SNP calling. Meanwhile, LoFreq does not provide zygosity in the result, which prohibited us from using "RTG vcfeval" for performance evaluation. Thus, we considered a true positive in LoFreq as having a matched truth record in (1) chromosome, (2) position, and (3) alternative allele. LoFreq requires base quality, and thus failed on the PacBio dataset, in which base quality is not provided. The results suggest that LoFreq is capable of SNP detection in SMS long reads. Unfortunately, we were unable to finish running Freebayes on both the PacBio dataset and the ONT dataset after they failed to complete on either dataset after running for 1 month. According to the percentage of genome covered with variant calls, we estimate that several months, 65 and 104 machine days on a latest 24-core machine, are required for a single PacBio and ONT dataset, respectively.

GIAB datasets were constructed from a consensus of multiple short-variant callers, and thus tend to bias toward easy regions that are accessible by these algorithms[23]. So, we next benchmarked the Syndip dataset, which is a recent benchmark dataset from the de novo PacBio assemblies of two homozygous human cell lines. As reported, the dataset provides a relatively more accurate and less biased estimate of small-variant-calling error rates in a realistic context[23]. The results are in Supplementary Table 3 and show that, when using Syndip variants for training, the performance of calling variants in both HG001 and HG002 at known variants remains as good as that previously reported. However, using the same model (Syndip), the performance dropped both at the Syndip known sites (excluding variants >4 bp, from 99.51% (HG001) to 98.52%) and for the whole genome (excluding variants >4 bp, from 94.88% (HG001) to 94.02%). The results support that Syndip contains variants that are harder to identify. To improve Clairvoyante's performance in the hard regions, we suggest users to also include Syndip for creating models.

**Potential novel variants unraveled by PacBio and ONT**. The truth SNPs and Indels provided by GIAB were intensively called and meticulously curated, and the accuracy and sensitivity of the GIAB datasets are unmatched. However, since the GIAB variants were generated without incorporating any SMS technology[17], it is possible that we can consummate GIAB by identifying variants not yet in GIAB, but specifically detected both by using the PacBio and the ONT data. For the HG001 sample (variants called in HG001 using a model trained on HG001), we extracted the so-called "false-positive" variants (identified genome-wide with a 0.2 alternative allele frequency cutoff) called in both the PacBio and ONT dataset. Then we calculated the geometric mean of the variant qualities of the two datasets, and we filtered the variants with a mean quality lower than 135 (calculated as the geometric mean of the two best variant quality cutoffs, 130 and 139). The resulting catalog of 3135 variants retained are listed in Supplementary Data 1. In total, 2732 are SNPs, 298 are deletions, and 105 are insertions. Among the SNPs, 1602 are transitions, and 1130 are transversions. The Ti/Tv ratio is ~1.42, which is substantially higher than random (0.5), suggesting a true biological origin. We manually inspected the top 10 variants in quality using IGV[24] to determine their authenticity (Fig. 1a and Supplementary Figure 1a-1i). Among the 10 variants, we have one convincing example at 2:163,811,179 (GRCh37) that GIAB has previously

missed (Supplementary Figure 1h). Another seven examples have weaker supports that need to be further validated using other orthogonal methods. Possible artifacts, including (1) 7:89,312,043 (Supplementary Figure 1g) have multiple SNPs in its vicinity, which is a typical sign of false alignment, (2) 1:566,371 (Supplementary Figure 1a) and 20:3,200,689 (Fig. 1a) are located in the middle of homopolymer repeats, which could be caused by misalignment, (3) X:143,214,235 (Supplementary Figure 1b) shows significant strand bias in Illumina data, and (4) X:140,640,513 (Supplementary Figure 1d), X:143,218,136 (Supplementary Figure 1e), and 9:113,964,088 (Supplementary Figure 1f) are potential heterozygous variants but with allele frequency notably deviated from 0.5. Two examples are because of the difference in representation—13:104,270,904 (Supplementary Figure 1c) and 10:65,260,789 (2i) have other GIAB truth variants in their 5-bp flanking regions. Manually inspecting all the 3135 variants is beyond the scope of this paper. However, our analysis suggests that SMS technologies, including both PacBio and ONT, can indeed generate some variants that are not identifiable by short-read sequencing. We advocate for additional efforts to look into these SMS-specific candidate variants systematically. The targets include not only shortlisting truth variants not yet in GIAB, but also new alignment and variant calling methods and algorithms to avoid detecting spurious variants in SMS data. Our analysis also serves as another piece of evidence that the GIAB datasets are of superior quality and are the enabler of machine-learning-based downstream applications such as Clairvoyante.

We also analyzed why the PacBio and ONT technologies cannot detect some variants. Figure 2 shows the number of known variants undetected by different combinations of sequencing technologies. We inspected the genome sequence immediately after the variants and found that among the 12,331 variants undetected by all three sequencing technologies, 3,289 (26.67%) are located in homopolymer runs, and 3,632 (29.45%) are located in short tandem repeats. Among the 178,331 variants that cannot be detected by PacBio and ONT, 102,840 (57.67%) are located in homopolymer runs, and 33,058 (18.54%) are located in short tandem repeats. For illustration, Fig. 1b–d depicted (b) a known variant in homopolymer runs undetected by all three sequencing technologies, (c) a known variant in short tandem repeats that cannot be detected with PacBio and ONT, and (d) a known variant flanked by random sequencing detected by all three sequencing technologies. It is a known problem that SMS technologies have significantly increased error rates at homopolymer runs and short tandem repeats[25]. Further improvements to the base-calling algorithm and sequencing chemistries will lead to raw reads with higher accuracy at these troublesome genome regions and hence, further decrease the number of known variants undetected by Clairvoyante.

## Discussion

In this paper, we presented Clairvoyante, a multitask convolutional deep neural network for variant calling using SMS. Its performance is on par with GATK UnifiedGenotyper on Illumina data and outperforms Nanopolish and DeepVariant on PacBio and ONT data. We analyzed the false-positive and false-negative variant calls in depth and found complex variants with multiple alternative alleles to be the dominant source of error in Clairvoyante. We further evaluated several different aspects of Clairvoyante to assess the quality of the design and how we can further improve its performance by training longer with a lower learning rate, combining multiple samples for training, or improving the input data quality. Our experiments on using Clairvoyante to call variants genome-wide suggested a range to search for the best

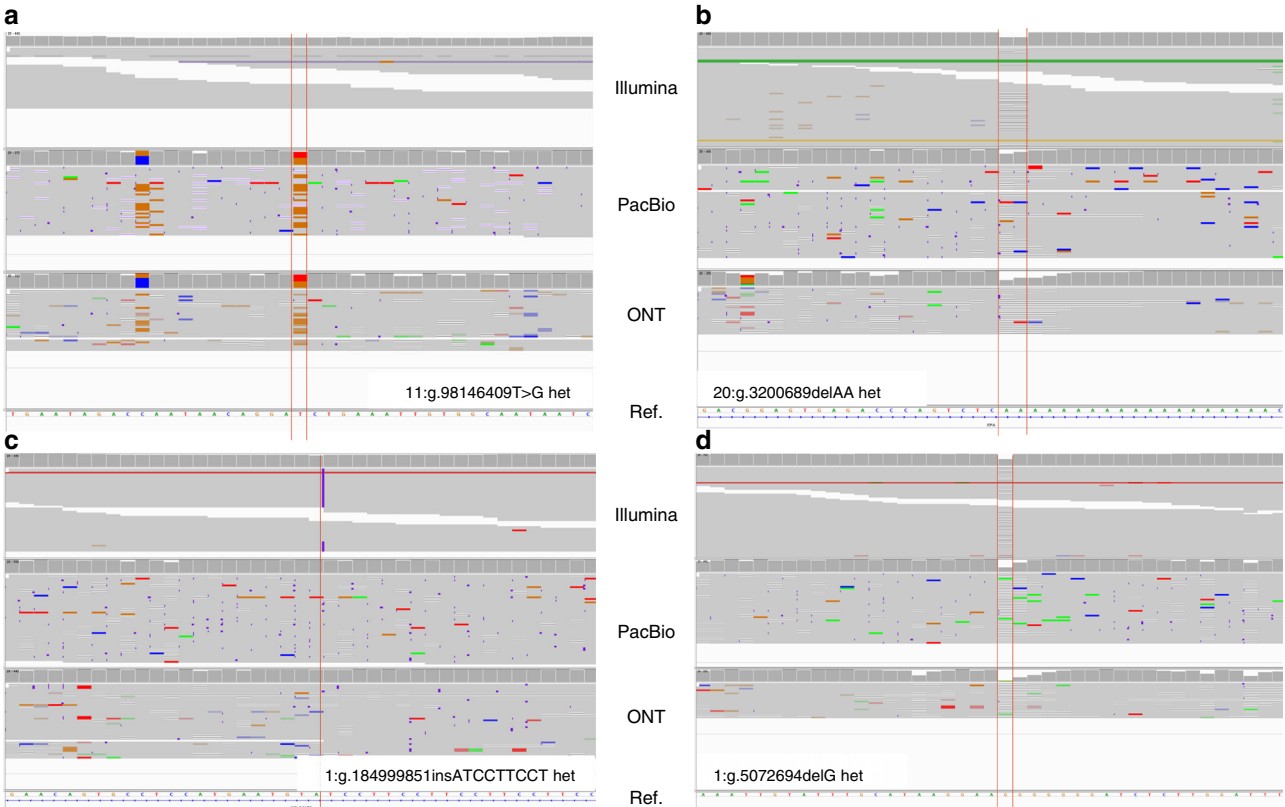

**Fig. 1** The IGV screen capture of the selected variants. **a** A heterozygote SNP from *T* to *G* at chromosome 11, position 98,146,409 called only in the PacBio and ONT data, **b** a heterozygote deletion *AA* at chromosome 20, position 3,200,689 not called in all three technologies, **c** a heterozygote insertion *ATCCTTCCT* at chromosome 1, position 184,999,851 called only in the Illumina data, and **d** a heterozygote deletion *G* at chromosome 1, position 5,072,694 called in all three technologies. The tracks from top to down show the alignments of the Illumina, PacBio, and ONT reads from HG001 aligned to the human reference GRCh37

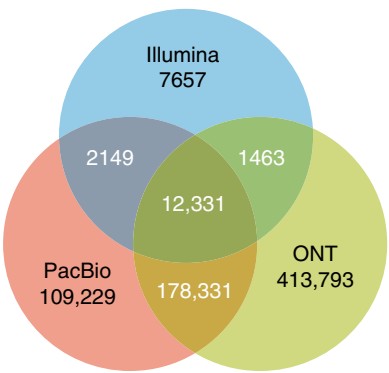

**Fig. 2** A Venn diagram that shows the number of undetected known variants by different sequencing technologies or combinations

alternative allele cutoff to balance the run time and recall for each sequencing technology. To the best of our knowledge, Clairvoyante is the first method for SMS to finish a whole-genome variant calling within 2 h on a single CPU-only server, while providing better precision and recall than other state-of-the-art variant callers such as Nanopolish. A deeper look into the so-called "false-positive" variant calls has identified 3135 variants in HG001 that are not yet in GIAB but detected by both PacBio and ONT independently. Inspecting 10 of these variants manually, we identified one strongly supported variant that should be included

by GIAB, seven variants with weak or uncertain supports that call for additional validation in a future study, and two variants actually exist in GIAB but with a different representation.

Clairvoyante relies on high-quality training samples to provide accurate and unbiased variant calling. This hinders Clairvoyante from being applied to completely novel sequencing technologies and chemistries, for which a high-quality sequencing dataset on standard samples such as GIAB has yet been produced. Nevertheless, with the increasing agreement for NA12878 as a gold-standard reference, this requirement seems to be quite manageable. Although Clairvoyante performed well on detecting SNPs, it still has a large room to be improved in detecting Indels, especially for ONT data, in which the Indel F1-score remains around 50%. To make the Indel results also practically usable, our target is to improve Clairvoyante further to reach an Indel F1-score over 80%. The current design of Clairvoyante ignores variants with two or more alternative alleles. Although the number of variants with two or more alternative alleles is small, a few thousands of the 3.5 M total sites, the design will be improved in the future to tackle this small but important group of variants. Due to the rareness of long indel variants for model training, Clairvoyante was set to provide the exact alternative allele only for indel variants ≤4 bp. The limitation can be lifted with more high-quality training samples available. The current Clairvoyante implementation also does not consider the base quality of the sequencing reads as Clairvoyante was targeting SMS, which does not have meaningful base quality values to improve the quality of variant calling. Nevertheless, Clairvoyante can be extended to

consider base quality by imposing it as a weight on depth or adding it as an additional tensor to the input. We do not suggest removing any alignment by their mapping quality because low-quality mappings will be learned by the Clairvoyante model to be unreliable. This provides valuable information about the trustworthiness of certain genomic regions. In future work, we plan to extend Clairvoyante to support somatic variant calling and trio-sample-based variant calling. Based on GIAB's high-confidence region lists for variant calling, we also plan on making PacBio-specific, and ONT-specific high-confidence region lists by further investigating the false-positive and false-negative variant calls made by Clairvoyante on the two technologies.

## Methods

**Overview**. In this section, we first introduce the DNA sequencing datasets of three different sequencing technologies: Illumina, PacBio, and ONT. We then formulate variant calling as a supervised machine-learning problem. Finally, we present Clairvoyante for this problem and explain the essential deep-learning techniques applied in Clairvoyante.

**Datasets**. While most of the variant calling in previous studies were done using a single computational algorithm on single-sequencing technology, the Genome-in-a-Bottle (GIAB) dataset[26] first published in 2014 has been an enabler of our work. The dataset provides high-confidence SNPs and indels for a standard reference sample HG001 (also referred to as NA12878) by integrating and arbitrating between 14 datasets from five sequencing and genotyping technologies, seven read mappers, and three variant callers. For our study, we used as our truth dataset the latest dataset version 3.3.2 for HG001 (Supplementary Note, Data Source, Truth Variants) that comprises 3,042,789 SNPs, 241,176 insertions, and 247,178 deletions for the GRCh38 reference genome, along with 3,209,315 SNPs, 225,097 insertions, and 245,552 deletions for GRCh37. The dataset also provides a list of regions that cover 83.8% and 90.8% of the GRCh38 and the GRCh37 reference genome, where variants were confidently genotyped. The GIAB extensive project[17] published in 2016 further introduced four standard samples, including the Ashkenazim Jewish sample HG002 we have used in this work, containing 3,077,510 SNPs, 249,492 insertions, and 256,137 deletions for GRCh37, 3,098,941 SNPs, 239,707 insertions, and 257,019 deletions for GRCh38. In total, 83.2% of the whole genome was marked as confident for both the GRCh37 and GRCh38.

Illumina Data: The Illumina data were produced by the National Institute of Standards and Technology (NIST) and Illumina[17]. Both the HG001 and HG002 datasets were generated on an Illumina HiSeq 2500 in Rapid Mode (v1) with 2 × 148-bp paired-end reads. Both have approximately 300× total coverage and were aligned to GRCh38 decoy version 1 using Novoalign version 3.02.07. In our study, we further downsampled the two datasets to 50× to match the available data coverage of the other two SMS technologies (Supplementary Note, Data Source, Illumina Data).

Pacific Bioscience (PacBio) Data: The PacBio data were produced by NIST and Mt. Sinai School of Medicine[17]. The HG001 dataset has 44× coverage, and the HG002 has 69×. Both datasets comprise 90% P6–C4 and 10% P5–C3 sequencing chemistry and have a sequence N50 length between 10k and 11 kbp. Reads were extracted from the downloaded alignments and aligned again to GRCh37 decoy version 5 using NGMLR[27] version 0.2.3 (Supplementary Note, Data Source, PacBio Data).

ONT Data: The Oxford Nanopore data were generated by the Nanopore WGS consortium[28]. Only data for sample HG001 are available to date, thus limiting the "cross sample variant calling evaluation" and "combined sampled training" on ONT data in the Results section. In our study, we used the 'rel3' release sequenced on the Oxford Nanopore MinION using 1D ligation kits (450 bp/s) and R9.4 chemistry. The release comprises 39 flowcells and 91.2 G bases, which is about 30× coverage. The reads were downloaded in raw fastq formatted and aligned to GRCh37 decoy version 5 using NGMLR[27] version 0.2.3 (Supplementary Note, Data Source, Oxford Nanopore Data).

**Variant calling as multitask regression and classification**. We model each variant with four categorical variables:

- $A \in \{A, C, G, T\}$ is the alternate base at a SNP, or the reference base otherwise
- $Z \in \{Homozygote, Heterozygote\}$ is the zygosity of the variant
- $T \in \{Reference, SNP, Insertion, Deletion\}$ is the variant type
- $L \in \{0, 1, 2, 3, 4, >4\}$ is the length of an INDEL, where ">4" represents a gap longer than 4 bp

For the truth data, each variable can be represented by a vector (i.e., a 1D tensor) using the one-hot or probability encoding, as is typically done in deep learning: $a_b = \Pr\{A = b\}$, $z_i = \delta(i, Z)$, $t_j = \delta(j, T)$, and $l_k = \delta(k, L)$, where $\delta(p, q)$ equals 1 if $p = q$, or 0 otherwise. The four vectors $(a, z, t, l)$ are the outputs of the network. $a_b$ is set to all zero for an insertion or deletion. In the current Clairvoyante

implementation, (1) multi-allelic SNPs are excluded from training, and (2) base quality is not used (see "Discussion" below for a rationale).

With deep learning, we seek a function $F: x \to (a, z, t, l)$ that minimizes the cost $C$:

$$C = \frac{1}{N} \sum_v \left( \sum_{b=1}^{4} \left( \hat{a}_b^{(v)} - a_b^{(v)} \right)^2 - \sum_{i=1}^{2} z_i^{(v)} \log \hat{z}_i^{(v)} - \sum_{j=1}^{4} t_j^{(v)} \log \hat{t}_j^{(v)} - \sum_{k=1}^{6} l_k^{(v)} \log \hat{l}_k^{(v)} \right),$$

where $v$ iterates through all variants and a variable with a caret indicates that it is an estimate from the network. Variable $x$ is the input of the network, and it can be of any shape and contains any information. Clairvoyante uses an $x$ that contains a summarized "piled-up" representation of read alignments. The details will be discussed in the next section named "Clairvoyante".

In our study, good performance implies that correct predictions could be made even when the evidence is marginal to distinguish a genuine variant from a nonvariant (reference) position. To achieve the goal, we paired each truth variant with two non-variants randomly sampled from the genome at all possible nonvariant and nonambiguous sites for model training. With about 3.5 M truth variants from the GIAB dataset, about 7 M nonvariants are added as samples for model training.

We randomly partitioned all samples into 90% for training and 10% for validation. We intentionally did not hold out any sample of the data for testing as other projects commonly do because, in our study, we can use an entirely different dataset for testing samples. For example, we can use the samples of HG002 to test against a model trained on HG001, and vice versa.

**Clairvoyante**. Clairvoyante is a multitask five-layer convolution neural network with the last two layers as feedforward layers (Fig. 3). The multitask neural network makes four groups of predictions on each input: (1) alternative alleles, (2) zygosity, (3) variant type, and (4) indel length. The predictions in groups 2–4 are mutually exclusive, while the predictions in group 1 are not. The alternative allele predictions are computed directly from the first fully connected layer (FC4), while the other three groups of predictions are computed from the second fully connected layer (FC5). The indel length prediction group has six possible outputs indicating an indel with a length between 0 and 3 bp or ≥4 bp of any unbounded length. The prediction limit on indel length is configurable in Clairvoyante and can be raised when more training data on longer indels could be provided. The Clairvoyante network is succinct and fine-tuned for the variant calling purpose. It contains only 1,631,496 parameters, which is about 13-times fewer than DeepVariant[10] using the Inception-v3 network architecture, which was originally designed for general-purpose image recognition. Additional details of Clairvoyante are introduced in the different subsections below.

For each input sample (truth or candidate variants), the overlapping sequencing read alignments are transformed into a multidimensional tensor $x$ of shape 33 by 4 by 4. The first dimension '33' corresponds to the position. The second dimension "4" corresponds to the count of A, C, G, or T on the sequencing reads, and the way of counting is subject to the third dimension. The third dimension "4" corresponds to four different ways of counting. In the first dimension, we added 16 flanking base pairs on both sides of a candidate (in total 33 bp), which we have measured to be sufficient to manifest background noise while providing a good computational efficiency. In the second dimension, we separated any counts into four bases. In the third dimension, we used four different ways of counting, generating four tensors of shape 33 by 4. The first tensor encodes the reference sequence and the number of reads supporting the reference alleles. The second, third, and fourth tensors use the relative count against the first tensor: the second tensor encodes the inserted sequences, the third tensor encodes the deleted base pairs, and the fourth tensor encodes alternative alleles. For an exact description of how $x$ is generated, refer to the pseudo code in "Supplementary Note, Pseudo code for generating the input". Figure 4 illustrates how the tensors can represent a SNP, an insertion, a deletion, and a nonvariant (reference), respectively. The nonvariant in Fig. 4 also depicts how the matrix will show background noise. A similar but simpler read alignment representation was proposed by Jason Chin[29] in mid-2017, the same time as we started developing Clairvoyante. Different from Chin's representation, ours decouples the substitution and insertion signal into separate arrays and allows us to precisely record the allele of the inserted sequence.

Our study used the widely adopted TensorFlow[30] as its primary programming framework. Using the 44× coverage HG001 PacBio dataset as an example, a near-optimal model can be trained in 3 h using the latest desktop GPU model nVidia GTX 1080 Ti. Using a trained model, about 2 h are needed to call variants genome-wide using a 2× 14-core CPU-only server (without GPU), and it takes only a few minutes to call variants at common variant sites or in an exome (>5000 candidate sites per second). Several techniques have been applied to minimize computational and memory consumption (see the Computational Performance section).

**Model initialization**. Weight initialization is important to stabilize the variances of activation and back-propagated gradients at the beginning of model training. We used a He initializer[31] to initialize the weights of hidden layers in Clairvoyante, as the He initializer is optimized for training extremely deep models using a rectified activation function directly from scratch. For each layer, the weight of each node is

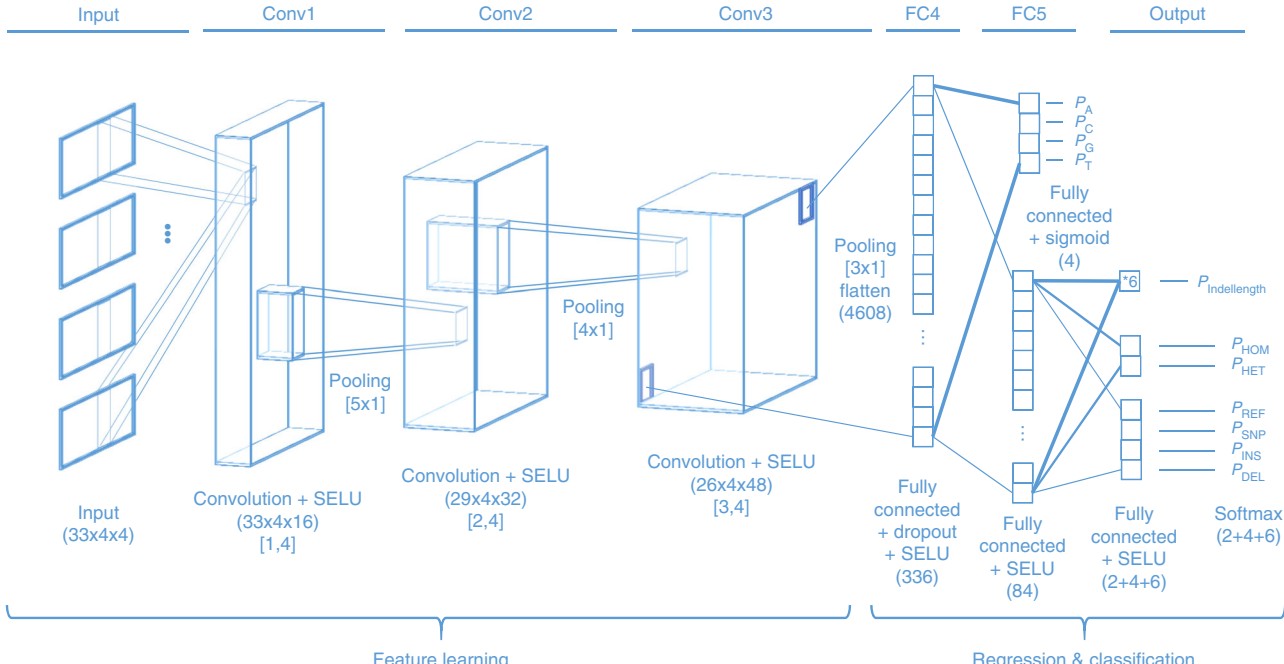

**Fig. 3** Clairvoyante network architecture and layer details. The descriptions under each layer, include (1) the layer's function; (2) the activation function used; (3) the dimension of the layer in parenthesis (input layer: height × width × arrays, convolution layer: height × width × filters, fully connected layer: nodes), and (4) kernel size in brackets (height × width)

sampled from a univariate normal distribution with $\sigma = 1 \div \sqrt[2]{d_i \div 2}$, where $d_i$ denote the number of in-degrees of the node.

**Activation function**. Batch normalization is a technique to ensure zero mean and unit variance in each hidden layer to avoid exploding or diminishing gradients during training. However, batch normalization has often been identified as a computational bottleneck in neural network training because computing the mean and the standard deviation of a layer is not only a dependent step, but also a reduction step that cannot be efficiently parallelized. To tackle this problem, we will use the new activation function called "Scaled Exponential Linear Units" (SELUs)[32], a variant of the rectified activation function. Different from a standard batch normalization approach that adds an implicit layer for the named purpose after each hidden layer, SELUs utilize the Banach fixed-point theorem to ensure convergence to zero mean and unit variance in each hidden layer without batch normalization.

**Optimizer and learning rate**. We used an Adam optimizer with default settings[33] to update the weights by adaptive node-specific learning rates, whereas setting a global learning rate only functions as setting an upper limit to the learning rates. This behavior allows Clairvoyante to remain at a higher learning rate for a longer time to speed up the training process.

Although the Adam optimizer performs learning rate decay intrinsically, we found that decreasing the global learning rate when the cost of the model in training plateaued can lead to a better model performance in our study. In Clairvoyante, we implemented two types of training modes. The fast training mode is an adaptive decay method that uses an initial learning rate at $1e^{-3}$, and decreases the learning rate by a factor of 0.1 when the validation rate goes up and down for five rounds and stops after two times of decay. A second nonstop training mode allows users to decide when to stop and continue using a lower learning rate.

**Dropout and L2 regularization**. Although more than 3 million labeled truth variants are available for training, the scarcity of some labels, especially variants with a long indel length, could fail the model training by overfitting to abundantly labeled data. To alleviate the class imbalance, we apply both dropout[34] and L2 regularization[35] techniques in our study. Dropout is a powerful regularization technique. During training, dropout randomly ignoring nodes in a layer with probability $p$, then sums up the activations of the remaining nodes and finally magnifies the sum by $1/p$. Then during testing, the algorithm sums up the activations of all nodes with no dropout. With probability $p$, the dropout technique is creating up to $1 \div (1 - p)^n$ possible subnetworks during the training. Therefore, dropout can be seen as dividing a network into subnetworks with reused nodes during training. However, for a layer with just enough nodes available, applying

dropout will require more nodes to be added, thus potentially increasing the time needed to train a model. In balance, we applied dropout only to the first fully connected layer (FC4) with $p = 0.5$, and L2 regularization to all the hidden layers in Clairvoyante. In practice, we set the lambda of L2 regularization the same as the learning rate.

**Visualization**. We created an interactive python notebook accessible within a web browser or a command line script for visualizing inputs and their corresponding node activations in hidden layers and output layers. Supplementary Figure 2 shows the input and node activations in all hidden layers and output layers of an A > G SNP variant in sample HG002 test against a model trained with samples from HG001 for a thousand epochs at $1e^{-3}$ learning rate. Each of the nodes can be considered as a feature deduced through a chain of nonlinear transformations of the read alignment input.

**Computational performance**. Making Clairvoyante a computationally efficient tool that can run on modern desktop and server computers with commodity configurations is one of our primary targets. Here, we introduce the two critical methods used for decreasing computational time and memory consumption.

Clairvoyante can be roughly divided into two groups of code, one is sample preparation (preprocessing and model training), and the second is sample evaluation (model evaluation and visualization). Model training runs efficiently because it invokes Tensorflow, which is maintained by a large developer community and has been intensively optimized with most of its performance critical code written in C, C++, or CUDA. Using the native python interpreter, sample preprocessing became the bottleneck, and the performance did not improve by using multi-threading due to the existence of global interpreter lock. We solved the problem by using Pypy[36], a Just-In-Time (JIT) compiler that performs as an alternative to the native python interpreter and requires no change to our code. In our study, Pypy sped up the sample preparation code by 5–10 times.

The memory consumption in model training was also a concern. For example, with a naive encoding, HG001 requires 40-GB memory to store the variant and non-variant samples, which could prevent effective GPU utilization. We observed that these samples are immutable and follow the "write once, read many" access pattern. Thus, we applied in-memory compression using the blosc[37] library with the lz4hc compression algorithm, which provides a high compression ratio, 100 MB/s compression rate, and an ultrafast decompression rate at 7 GB/s. Our benchmarks show that applying in-memory compression does not impact the speed but decreased the memory consumption by five times.

**Code availability**. Clairvoyante is available open-source at https://github.com/aquaskyline/Clairvoyante.

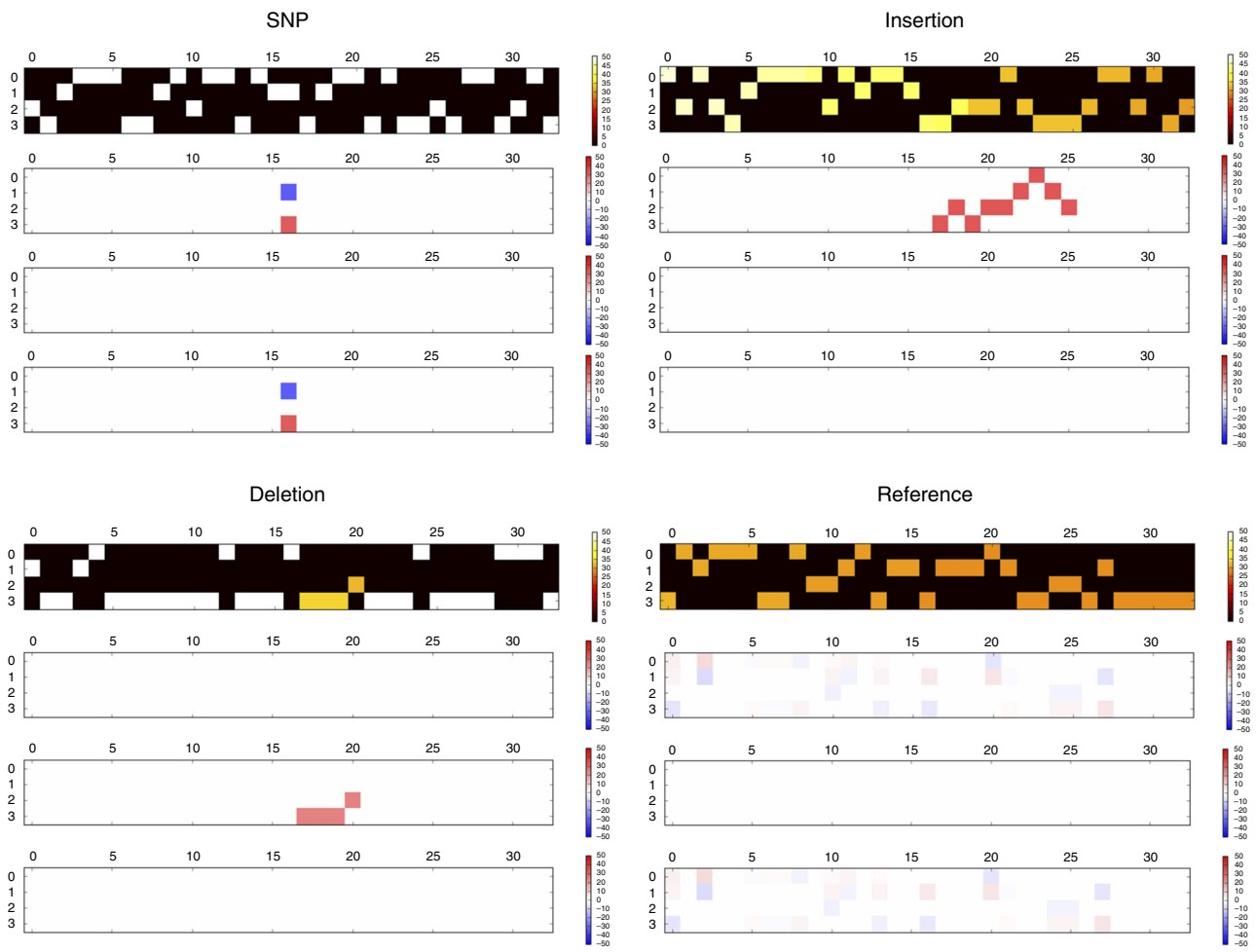

**Fig. 4** Selected illustrations of how Clairvoyante represents the three common types of a small variant, and a nonvariant. The figure includes: (top left) a *C* > *G* SNP, (top right) a 9-bp insertion, (bottom left) a 4-bp deletion, and (bottom right) a nonvariant with a reference allele. The color intensity represents the strength of a certain variant signal. The SNP insertion and deletion examples are ideal with almost zero-background noise. The nonvariant example illustrates how the background noises look like when not mingled with any variant signal

**Reporting summary**. Further information on experimental design is available in the Nature Research Reporting Summary linked to this article.

## Data availability
The authors declare that all data supporting the findings of this study are available by the links within the paper and its Supplementary information files. All other relevant data are available upon request.

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

## Acknowledgments

We thank Guangyu Yang for adding code to Clairvoyante to enable visualization using TensorBoard. We thank Chi-Man Liu and Yifan Zhang for their constructive comments and benchmarking Nanopolish. R.L. was supported by the General Research Fund No. 27204518, HKSAR. T.L. was partially supported by Innovative and Technology Fund ITS/331/17FP from the Innovation and Technology Commission, HKSAR. This work was also supported, in part, by awards from the National Science Foundation (DBI-1350041) and the National Institutes of Health (R01-HG006677 and UM1-HG008898).

## Author contributions

R.L. and M.S. conceived the study. R.L., F.S., T.L. and M.S. analyzed the data and wrote the paper.

## Additional information

**Competing interests:** The authors declare no competing interests.

