## [Peer Review File · Nature Communications]

Reviewers' comments:

Reviewer #1 (Remarks to the Author):

In their paper Luo et al present a new method Clairvoyante for variant calling that supports Illumina, Pacbio and Oxford Nanopore technologies. Inspired by recent Google's DeepVariant package the authors use deep convolutional neural networks. Unlike DeepVariant, the authors do not convert aligned reads to pictures. The authors present the method, compare it with GATK for Illumina reads and Nanopolish for nanopore reads and analyzed false positives and false negatives. In the last part of the paper the focus is on the identification of previously unknown variants. In accordance with results achieved on both Pacbio and nanopore data the authors propose 3,135 new variants in HG001.

The source code of the Clairvoyante package, is available on github. The instructions and examples are well described and elaborated.

Although the paper is well written and organized, it lacks the comprehensive comparison with the state-of-the-methods. The authors evaluated GATK for Illumina reads and nanopolish for nanopore reads, but ie. they didn't compare their method with any other variant caller that supports PacBio reads.

In their recent paper Sadman et al ("Evaluating Variant Calling Tools for Non-Matched Next-Generation Sequencing Data") compared several variant calling methods for NGS data. Among others they highlighted VarDict (the best performance) FreeBayes and LoFreq. Unfortunately, none of them are evaluated in this paper. In addition, to the best of my knowledge, all three of them support PacBio reads and some of them have been used for nanopore reads as well:

- LoFreq in Sovic et al - Fast and sensitive mapping of nanopore sequencing reads with GraphMap
- Freebayes in Miten et al - Nanopore sequencing and assembly of a human genome with ultra-long reads.

The Clairvoyante variant caller, described in the paper, seems sound and is well presented. However, the readers should be convinced that it performs better than existing methods.

Minor comment:

The Figure 2 should be of better quality and more clearly described.

Reviewer #2 (Remarks to the Author):

The authors have developed a variant calling method called Clairvoyante. Clairvoyante is using deep learning, particularly the convolutional neural network (CNN), to detect germline variants. The method detects SNP/Indel with zygosity. The method reported to have achieved an accuracy (F1) of over 98% for short reads and from 77%-93% for long reads with three different sequencing technologies, along with few thousands of high-confidence variants that were long-read-technology specific. The reported execution time was less than a couple of hours.

In terms of novelty of the approach, this method is using Deep Convolutional Neural Network to call genetic variations, which is same as DeepVariant; however, DeepVariant predicts variants by learning the likelihoods between images of reads pileups around putative variants sites and ground-truth genotype calls. This method is taking a base-pair and counting approach, which also predicts the the alternative allele and variant type directly from the network. This method is different in terms of implementation, architecture and capability.

For testing and validation, the paper said “We intentionally did not hold out any sample of the data for testing as other projects commonly do because, in our study, we can use an entirely different dataset for testing samples. For example, we can use the samples of HG002 to test against a model trained on HG001, and vice versa.” It also said “Only data for sample HG001 are available to date, thus limiting the cross sample variant calling evaluation and combined sampled training on ONT data in the Result section.” If that’s the case, why the authors chose to partition all samples into 90% for training and 10% for validation and did not hold out any sample for the ONT data for testing? As the tuning is based on the validation set, it would be more valid to partition for test set using the HG001 for the ONT case.

As to the choice of variant callers to compare, the authors have chosen GATK Unified Genotyper (UG), GATK HaplotypeCaller (HC), and Nanopolish. It was unclear why GATK UG was chosen as its almost deprecated, which clearly will show sub-optimal results and GATK HC is the one being actively maintained. The results from GATK UG certainly contrasts the results from the paper, but could be a confusing baseline for other readers who are not so familiar with the details.

Both GATK UG and HC were not long read callers, they were designed mostly for Illumina reads. The benchmarking should seek for a way to use a real long read caller designed for high error rate or more designed for the sequencing technology, if this paper is focused on and titled with “Single Molecule Sequencing”. The paper did point out they tried GenomicConsensus and paftools but couldn’t get it to work, but the authors should resolve the problems and provide a fair comparison to the readers. For example, can the authors modify the downloaded BAM such that it will work for Quiver (GenomicConsensus)? e.g. can the authors start from H5 and use Blasr/Quiver? even Nanopolish is slow, is it impossible to finish the whole genome given the compute environment of the authors?

Since DeepVariant is currently the state-of-the-art in using Deep Learning to predict variants, it is critical for this paper to have a direct comparison to the results of DeepVariant, e.g. running the latest version of DeepVariant on the same datasets, in addition to comparing to a short read caller on long reads. Both direct comparison for accuracy and speed with the same/similar compute resources would be very beneficial to the readers.

The benchmarking should also separate the SNP and INDEL accuracies clearly, as INDEL usually exhibits more difficulties in detection. E.g. in the precisionFDA truth challenge, DeepVariant scored 99.96% on SNP and only 98.98% on INDELS.

Overall, the benchmarking main tables should also have an F1 score column for Genome Wide calls in addition to known sites.

Reviewer #3 (Remarks to the Author):

As my comments involve many math notations, I am submitting them as a PDF.

Luo et al developed Clairvoyante, a software package that employs deep learning models to genotype and to call SNPs and small INDELS from read alignments. Along with Google’s deepVariant, which is unpublished, Clairvoyante is one of the first genotypers/variant callers that explore deep learning and work reasonably well with PacBio and Oxford Nanopore data. Overall, this manuscript clearly presents the results and shows convincing performance. Critical observations are as follows.

1. I am unable to run Clairvoyante due to the older machine I have. The culprit is the combination of pypy and blosc. I can install all relevant python libraries with conda. However, Clairvoyante seems to require pypy to run; otherwise I will get a runtime error. I need to install blosc with pypy, which leads a compiling error due to the lack of AVX2 on my machine. I hope the authors may consider to make Clairvoyante via Bioconda. I know CPython is slow but at least it is usable.
2. About the practical use cases of Clairvoyante. Table 7 suggests that the long-read accuracy is much lower than the short-read accuracy and that for short reads Clairvoyante is not as accurate as GATK (see also point 4 below). Then why would we use Clairvoyante? In my opinion, the authors should demonstrate applications where Clairvoyante clearly beats GATK. One such application is variant filtering: we take raw GATK calls as input and identify good variants. This strategy combines deep learning with the power of the GATK algorithm, and thus has the potential to outperform GATK. I think the authors should try this. It is not hard with their current framework.
3. It is not clear to me what “known variant sites” mean in Table 2 through 4. It could have two distinct meanings: 1) sites known to be polymorphic in one sample; 2) sites known to be polymorphic in the much larger population. If I am right, “known variant sites” in the manuscript refer to 1). If this is the case, I am afraid that evaluating on sample-specific variants has little to do with real applications. I think Table 2 through 4 are misleading and should be moved to the supplementary.

The right way to evaluate genotyping accuracy is to take the 1000g (or gnomAD, TopMed, HRC etc) polymorphic sites, use Clairvoyante to call at those sites and then compare to the GIAB truth. The authors should take the intersection of GIAB and 1000g as the true variant set, to avoid classifying GIAB sites missing from 1000g as FNs. I think having this evaluation would be good given that calling small variants from long reads is error prone. I hope, though I wouldn’t demand, the authors to add this evaluation. Point 2 is more important.

4. Table 7 evaluates Clairvoyante in the discovery mode. I think the GATK accuracy here is too low. On GIAB, GATK typically achieves an F1 of 0.995–0.999 (see the precisionFDA results: <https://precision.fda.gov/challenges/truth/results-explore>; I got 0.997 with GATK on a different HG001 BAM). The authors’ results 0.989 are over twice as worse (0.005 vs 0.011). The authors should investigate why this is the case.
5. The method description and math notations need improvements. First, note that the space of all $m \times n$ matrices (i.e. 2-D tensors) is denoted by $\mathbb{R}^{m \times n}$, so similarly 3-D tensor space is denoted by $\mathbb{R}^{m \times n \times p}$. The superscript indicates dimensions. Second, with authors’ notations \mathbb{R}_w^V , the superscript is irrelevant and should be removed. Third, variables like z_i are defined, but z_{ix} etc are not defined. The following is how I would describe the model (*I don’t understand how the input tensor is constructed; please explain more clearly*):

We model each variant with four categorical variables:

- $A \in \{\text{A, C, G, T}\}$ is the alternate base at a SNP, or the reference base otherwise.
- $Z \in \{\text{homozygote, heterzygote}\}$ is the zygosity of the variant.
- $T \in \{\text{reference, SNP, insertion, delete}\}$ is the variant class.
- $L \in \{0, 1, 2, 3, 4, \geq 5\}$ is the length of an INDEL, where ‘ ≥ 5 ’ represents a gap 5bp or longer.

For the truth data, each variable can be represented by a vector (i.e. 1-D tensor) using the one-hot or probability encoding, as is typically done in deep learning: $a_b = \Pr\{A = b\}$, $z_i = \delta(i, Z)$, $t_j = \delta(j, T)$ and $l_k = \delta(k, L)$, where $\delta(i, j)$ equals one if $i = j$, or 0 otherwise. The four vectors (a, z, t, l) are the outputs of the network.

The input of the network encodes allele counts. (*reviewer comment: the authors said “the second tensor encodes the inserted sequences”, but in authors’ formulation, there is only one input tensor x . To describe a tensor, they need to explain what each dimension represents. I understand in $33 \times 4 \times 4$ ‘33’ corresponds to position. I guess the first ‘4’ corresponds to A/C/G/T on the reference (or on read?) and the second ‘4’ to the four allele types as variable T above. However, I still don’t understand how the relative counts are derived – what’s the count at (A,insertion)? The authors should show an example about how these $4 \times 4 = 16$ numbers are calculated in exact. How to translate real data into tensors is one of the most important aspects in deep learning.*)

With deep learning, we seek a function $F : x \mapsto (a, z, t, l)$ that minimizes the cost:

$$C = \frac{1}{N} \sum_v \left(\sum_{b=1}^4 (\hat{a}_b^{(v)} - a_b^{(v)})^2 - \sum_{i=1}^2 z_i^{(v)} \log \hat{z}_i^{(v)} - \sum_{j=1}^4 t_j^{(v)} \log \hat{t}_j^{(v)} - \sum_{k=1}^6 l_k^{(v)} \log \hat{l}_k^{(v)} \right)$$

where v iterates through all variants and a variable with a caret indicates it is an estimate from the network.

The following are minor comments:

6. I believe the method description also contains minor errors or subtleties. In no particular order:

- In authors’ notation, a_i “denotes the probability of four possible reference and alternative alleles”. How to set a_i given an INDEL and a multi-allelic SNP?
- Does the model use base quality?
- Cross-entropy takes the form of $\sum_i z_i \log \hat{z}_i$, where z_i is the truth. In authors’ equation, z_i and \hat{z}_i are flipped.

7. It would be good to evaluate using the syndip dataset (<https://github.com/lh3/CHM-eval>). GIAB is highly biased towards easy regions and is not a good representative of real applications.

8. P2L64 (page 2, line 64). The initial deepVariant had to encode alignment as an image as it was using Google’s DisBelief framework which only takes images as input. However, the latest deepVariant is built on top of TensorFlow as well. It doesn’t have the old limitation any more.

9. P9L327. It is not necessary to evaluate paftools from minimap2. It was not designed to call variants from raw reads.

10. P11L396. I am not sure how numbers are added up. The authors chose 100 FPs, of which 71 FPs are caused by multi-allelic sites. Then later the authors claimed only 12 FPs among the 100 FPs are true Clairvoyante errors. Wouldn’t those 71 FPs are all Clairvoyante errors, too?

It is worth noting that variant callers are responsible for identifying alignment artifacts, which are often associated with distinct features (e.g. higher depth and higher variant density). Failing to detect alignment errors is still an error. GIAB is highly accurate. Based on the precision of Clairvoyante, I don’t think 20% of differences could be GIAB errors. In addition, GIAB was constructed from a variety of data sources. Variants ambiguous in Illumina data may be clear in data sequenced with other technologies. How to ‘validate’ variants in IGV is also tricky. RTG’s vcfEval may do sophisticated variant realignment, which may not be obvious to eyes. Overall, I think the discussion in this paragraph is too defensive. GIAB has done a phenomenal job on eliminating errors. If the authors want to investigate GIAB errors, they have to look at the results of multiple other variant callers, other short-read callsets and other technologies.

11. P16L552. 3,135 missing variants in GIAB is an overestimate. There are at least two errors in the authors’ analysis: GIAB contains variants at 13:104270904 and 10:65260789. Most other examples also look dubious. For example, on a different dataset, there are multiple 2bp deletions at 20:3,200,689 in Illumina data. 7:89312043 has multiple SNPs in a small region. This is a typical sign of false alignment. 1:566371 is located in the middle of repeats and could be caused by misalignment. X:143214235 shows

extreme strand bias in Illumina data. I need to see the strands of long-read data and mapping quality to tell if it has a similar issue. The only convincing example where GIAB is wrong is 2:163811179. To this end, my observation strongly contradicts the authors' claim that "all ten variants ... are strongly supported by the data." Overall, the authors should weaken their claim in this paragraph. GIAB may miss variants, but there should be much fewer than 3,135.

12. At last, the authors should give deepVariants more credits. DeepVariants uses a much more sophisticated, though much slower, model and significantly outperforms GATK on the GIAB benchmark, which has been independently confirmed by multiple groups. DeepVariants also works for PacBio data. I know deepVariants is not published yet, but I hope the authors may consider to evaluate it as well. A good paper should consider all available tools, peer-reviewed or not.

Reviewer: Heng Li (hengli@broadinstitute.org)

Reviewer: 1

In their paper Luo et al present a new method Clairvoyante for variant calling that supports Illumina, Pacbio and Oxford Nanopore technologies. Inspired by recent Google's DeepVariant package the authors use deep convolutional neural networks. Unlike DeepVariant, the authors do not convert aligned reads to pictures. The authors present the method, compare it with GATK for Illumina reads and Nanopolish for nanopore reads and analyzed false positives and false negatives. In the last part of the paper the focus is on the identification of previously unknown variants. In accordance with results achieved on both Pacbio and nanopore data the authors propose 3,135 new variants in HG001.

The source code of the Clairvoyante package, is available on github. The instructions and examples are well described and elaborated.

Although the paper is well written and organized, it lacks the comprehensive comparison with the state-of-the-methods. The authors evaluated GATK for Illumina reads and nanopolish for nanopore reads, but ie. they didn't compare their method with any other variant caller that supports PacBio reads. In their recent paper Sadman et al ("Evaluating Variant Calling Tools for Non-Matched Next-Generation Sequencing Data") compared several variant calling methods for NGS data. Among others they highlighted VarDict (the best performance) FreeBayes and LoFreq. Unfortunately, none of them are evaluated in this paper. In addition, to the best of my knowledge, all three of them support PacBio reads and some of them have been used for nanopore reads as well:

- LoFreq in Sovic et al - Fast and sensitive mapping of nanopore sequencing reads with GraphMap
- Freebayes in Miten et al - Nanopore sequencing and assembly of a human genome with ultra-long reads.

The Clairvoyante variant caller, described in the paper, seems sound and is well presented. However, the readers should be convinced that it performs better than existing methods.

RE: We thank the reviewer for their suggestion on presenting a more comprehensive comparison, especially with the three other tools VarDict, LoFreq, and FreeBayes. While the paper suggested by the reviewer titled "Evaluating Variant Calling Tools for Non-Matched Next-Generation Sequencing Data" by Sadman et al. was focused on evaluating the performance of calling somatic variant, we took the paper as a guide of best practice and benchmarked the three tools' performance on detecting germline variant. As suggested by another reviewer, we have also benchmarked DeepVariant, which is believed to be the first deep neural network based variant caller designed for calling germline variants in Illumina data. The results were added to the manuscript as an additional section named "Benchmarks of other state-of-the-art variant callers". The results (VCF files) of the additional benchmarks are available at the link: <https://goo.gl/s25wCF>, except for those failed experiments due to reasons such as running for too long, or lack of base quality but required by the caller. The commands used are also presented in the Supplementary Note.

Unlike the Illumina or Oxford Nanopore data, on which multiple variant callers ran successfully, we were unable to finish running any caller on the Pacbio data due to two reasons. The first reason is, Vardict, LoFreq, and DeepVariant require base quality, which is not provided in the GIAB HG001 PacBio dataset (available at ftp://ftp-trace.ncbi.nlm.nih.gov/giab/ftp/data/NA12878/NA12878_PacBio_MtSinai/). It would also not be fair to add in "dummy" quality values such as QV8 or QV9 (a typical PacBio base quality) because the other methods expect the quality values to contain meaningful information. We searched for available PacBio datasets with known variants and contain base quality but failed, but we found that most of the PacBio alignments are without base quality probably because none of PacBio's proprietary SMRT Analysis tools use or need them. The second reason is Freebayes requires more than two months to finish running on the PacBio dataset: We ran Freebayes for a month before stopping, and using the percentage finished, we estimate 65 days for running Freebayes on the Pacbio dataset. We suspect the extreme running time is because Freebayes considers every site with at least one discrepant read, and with PacBio data, nearly every position in the genome will need to be evaluated. The

results, however, have again highlighted Clairvoyante's value on calling small variants in PacBio datasets effectively and efficiently.

Minor comment:

The Figure 2 should be of better quality and more clearly described.

RE: We have added a high-resolution Figure 2 (with 2385 × 1792 pixels) and updated the description to make it clearer.

Reviewer: 2

The authors have developed a variant calling method called Clairvoyante. Clairvoyante is using deep learning, particularly the convolutional neural network (CNN), to detect germline variants. The method detects SNP/Indel with zygosity. The method reported to have achieved an accuracy (F1) of over 98% for short reads and from 77%-93% for long reads with three different sequencing technologies, along with few thousands of high-confidence variants that were long-read-technology specific. The reported execution time was less than a couple of hours.

In terms of novelty of the approach, this method is using Deep Convolutional Neural Network to call genetic variations, which is same as DeepVariant; however, DeepVariant predicts variants by learning the likelihoods between images of reads pileups around putative variants sites and ground-truth genotype calls. This method is taking a base-pair and counting approach, which also predicts the alternative allele and variant type directly from the network. This method is different in terms of implementation, architecture and capability.

For testing and validation, the paper said "We intentionally did not hold out any sample of the data for testing as other projects commonly do because, in our study, we can use an entirely different dataset for testing samples. For example, we can use the samples of HG002 to test against a model trained on HG001, and vice versa." It also said "Only data for sample HG001 are available to date, thus limiting the cross sample variant calling evaluation and combined sampled training on ONT data in the Result section." If that's the case, why the authors chose to partition all samples into 90% for training and 10% for validation and did not hold out any sample for the ONT data for testing? As the tuning is based on the validation set, it would be more valid to partition for test set using the HG001 for the ONT case.

RE: We thank the reviewer for their suggestions. Using HG001, we trained a model without chromosome 1 and benchmarked the performance of the trained model on chromosome 1. Compared to the previous results that called variants on all chromosomes against a model trained on the same chromosomes, the new results are very similar and slightly better. We have added the new results to Table 4.

As to the choice of variant callers to compare, the authors have chosen GATK Unified Genotyper (UG), GATK HaplotypeCaller (HC), and Nanopolish. It was unclear why GATK UG was chosen as its almost deprecated, which clearly will show sub-optimal results and GATK HC is the one being actively maintained. The results from GATK UG certainly contrasts the results from the paper, but could be a confusing baseline for other readers who are not so familiar with the details.

RE: To avoid misunderstanding and to help the readers to value the GATK UG results appropriately, we have added the following descriptions to the revision: "Noteworthy, GATK UnifiedGenotyper was superseded by GATK HaplotypeCaller, thus for Illumina data, we should refer to the results of HaplotypeCaller as the true performance of GATK. However, our benchmarks show that UnifiedGenotyper performed better than HaplotypeCaller on the PacBio and ONT data, thus we also benchmarked UnifiedGenotyper for all three technologies for users to make parallel comparisons."

Both GATK UG and HC were not long read callers, they were designed mostly for Illumina reads. The benchmarking should seek for a way to use a real long read caller designed for high error rate or more designed for the sequencing technology, if this paper is focused on and titled with "Single Molecule Sequencing". The paper did point out they tried GenomicConsensus and paftools but couldn't get it to work, but the authors should resolve the problems and provide a fair comparison to the readers. For example, can the authors modify the downloaded BAM such that it will work for Quiver (GenomicConsensus)? e.g. can the authors start from H5 and use Blasr/Quiver? even Nanopolish is slow, is it impossible to finish the whole genome given the compute environment of the authors?

RE: We have added Nanopolish's result on the whole genome to the revision. It ran for 40 days but was a worthwhile test because the whole genome result is noticeably better than the chr19 result (F1-score 90.47% against 88.06%). We removed pafitools according to reviewer 3's comments, as he is the author of pafitools. We tried to use GenomicConsensus with our BAM file, but abandoned the attempt after two days as we saw no hope of getting any reasonable result. We wrote to the authors at PacBio but got a negative reply that we should start with the BAM file that was generated by PacBio's proprietary pipeline (not provided by GIAB, and generally not available). Fortunately, we were able to benchmark DeepVariant on both the Illumina data and ONT data. The new benchmarks are in the new section named "Benchmarks of other state-of-the-art variant callers" and they support our conclusion that Clairvoyante has the best speed and overall performance to date for long read data. But again, we were unable to run DeepVariant on the PacBio data because DeepVariant requires base quality that the GIAB HG001 PacBio dataset doesn't have, which is a known "problem" of most of the PacBio datasets publicly available. We designed Clairvoyante with this known "problem" in mind, thus not requiring base quality and can be applied to known Single Molecule Sequencing technologies smoothly.

Since DeepVariant is currently the state-of-the-art in using Deep Learning to predict variants, it is critical for this paper to have a direct comparison to the results of DeepVariant, e.g. running the latest version of DeepVariant on the same datasets, in addition to comparing to a short read caller on long reads. Both direct comparison for accuracy and speed with the same/similar compute resources would be very beneficial to the readers.

RE: We added the benchmarks of the latest version of DeepVariant on both the Illumina data and ONT data to the new section in the revision named "Benchmarks of other state-of-the-art variant callers".

The benchmarking should also separate the SNP and INDEL accuracies clearly, as INDEL usually exhibits more difficulties in detection. E.g. in the precisionFDA truth challenge, DeepVariant scored 99.96% on SNP and only 98.98% on INDELS.

RE: In the revision, for all benchmarks, we have shown the Precision, Recall and F1-score of Overall, SNP, and Indel, respectively.

Overall, the benchmarking main tables should also have an F1 score column for Genome Wide calls in addition to known sites.

RE: We understood the reviewer and readers' need for comparing the two types of benchmarks more efficiently. We have added the Supplementary Table 3 titled "Summary of the best precision, recall, and F1-score of the benchmarks at known sites and on whole-genome" in the revision.

Reviewer: 3 (Heng Li, hengli@broadinstitute.org)

Luo et al developed Clairvoyante, a software package that employs deep learning models to genotype and to call SNPs and small INDELS from read alignments. Along with Google's deepVariant, which is unpublished, Clairvoyante is one of the first genotypers/variant callers that explore deep learning and work reasonably well with PacBio and Oxford Nanopore data. Overall, this manuscript clearly presents the results and shows convincing performance. Critical observations are as follows.

1. I am unable to run Clairvoyante due to the older machine I have. The culprit is the combination of pypy and blosc. I can install all relevant python libraries with conda. However, Clairvoyante seems to require pypy to run; otherwise I will get a runtime error. I need to install blosc with pypy, which leads a compiling error due to the lack of AVX2 on my machine. I hope the authors may consider to make Clairvoyante via Bioconda. I know CPython is slow but at least it is usable.

RE: We thank you for your thorough and constructive reviews and suggestions. We have added Clairvoyante to bioconda and add a section named "Using bioconda" to the README in GitHub for how to install Clairvoyante through bioconda. Even without using bioconda, it is not necessary to install blosc for pypy now, only intervaltree is required by pypy in the new version.

2. About the practical use cases of Clairvoyante. Table 7 suggests that the long-read accuracy is much lower than the short-read accuracy and that for short reads Clairvoyante is not as accurate as GATK (see also point 4 below). Then why would we use Clairvoyante? In my opinion, the authors should demonstrate

applications where Clairvoyante clearly beats GATK. One such application is variant filtering: we take raw GATK calls as input and identify good variants. This strategy combines deep learning with the power of the GATK algorithm, and thus has the potential to outperform GATK. I think the authors should try this. It is not hard with their current framework.

RE: We implemented the proposed application but decided that since the focus of this paper was around single molecule sequencing present those results in a separate preprint. For this manuscript we have added the following description to the revision: "We realized, on Illumina data, Clairvoyante is not performing on-par with the state-of-the-art GATK HaplotypeCaller, which was intensively optimized for Illumina data. However, as Clairvoyante uses an entirely different algorithm from GATK, we believe it can be used as an orthogonal method for filtering or validating GATK's results to increase GATK's accuracy further, emulating how geneticists might manually validate a variant using a genome browser. We implemented the method and named it Skyhawk. It repurposed Clairvoyante's neural network to work on the GATK's variants, and gives them another quality score in addition to the existing one by GATK to streamline manual review. More details are available in Skyhawk's preprint (<https://doi.org/10.1101/311985>). With the success of developing Skyhawk, we expect to see in the future, more applications would be developed upon Clairvoyante's network architecture."

3. It is not clear to me what "known variant sites" mean in Table 2 through 4. It could have two distinct meanings: 1) sites known to be polymorphic in one sample; 2) sites known to be polymorphic in the much larger population. If I am right, "known variant sites" in the manuscript refer to 1). If this is the case, I am afraid that evaluating on sample-specific variants has little to do with real applications. I think Table 2 through 4 are misleading and should be moved to the supplementary.

The right way to evaluate genotyping accuracy is to take the 1000g (or gnomAD, TopMed, HRC etc) polymorphic sites, use Clairvoyante to call at those sites and then compare to the GIAB truth. The authors should take the intersection of GIAB and 1000g as the true variant set, to avoid classifying GIAB sites missing from 1000g as FNs. I think having this evaluation would be good given that calling small variants from long reads is error prone. I hope, though I wouldn't demand, the authors to add this evaluation. Point 2 is more important.

RE: We have added the following discussion to the revision to tackle the concerns and avoid misleading the readers: "Noteworthy, the benchmarks at known GIAB truth variant sites 1) provided a purer view of how sequencing technologies perform differently with Clairvoyante and other tools in the "less complicated" genome regions, which in turn 2) enables the detailed assessments of Clairvoyante including testing for overfitting, higher data quality and network capacity. The benchmarks also 3) support the expected performance of Clairvoyante on a typical precision medicine application that only tens to hundreds of clinically relevant or actionable variants are being genotyped. This is especially true now that single molecule sequencing is starting to become more widely used for clinical diagnosis of structural variations, but at the same time, doctors also want to know if there are any actionable or incidental small variants without additional short read (Illumina) sequencing (author's comment: indeed, this is a project we are working on with a local hospital in Hong Kong). However, this section does not show Clairvoyante's performance on genome-wide variant identification, which is a more common application of a variant caller. Please refer to the section named "Genome-wide variant identification" for Clairvoyante's performance genome-wide." We tested using GIAB and dbSNPv146 as the true variants for benchmarking. As expected, the performance drops between using GIAB only and using the whole-genome. However, as scenario 1 evaluates the benchmarks at the known GIAB sites to support the detailed assessments, and scenario 2 evaluates whole-genome variant identification results to reflect Clairvoyante's real-life performance, we would like to keep both the "GIAB known variants" and "genome-wide" benchmarks in the main text.

4. Table 7 evaluates Clairvoyante in the discovery mode. I think the GATK accuracy here is too low. On GIAB, GATK typically achieves an F1 of 0.995–0.999 (see the precisionFDA results: <https://precision.fda.gov/challenges/truth/results-explore>; I got 0.997 with GATK on a different HG001 BAM). The authors' results 0.989 are over twice as worse (0.005 vs 0.011). The authors should investigate why this is the case.

RE: We have updated the GATK results in all tables and the relevant descriptions. The F1-score we have achieved on HG001 is 0.9976. The BAM files we have used for analysis are available at <http://www.bio8.cs.hku.hk/clairvoyante/bamUsed/> for verification. The previous numbers were generated with indels >4bp removed from both the results and baselines, we did it so as to unify with the benchmarks of Clairvoyante. However, in the new revision, we used the full callset of GATK as well as other variant callers against the full baseline to reflect their real performance.

5. The method description and math notations need improvements. First, note that the space of all $m \times n$ matrices (i.e. 2-D tensors) is denoted by $R^{m \times n}$, so similarly 3-D tensor space is denoted by $R^{m \times n \times p}$. The superscript indicates dimensions. Second, with authors' notations R^V_w , the superscript is irrelevant and should be removed. Third, variables like z_i are defined, but z_{ix} etc are not defined. The following is how I would describe the model (I don't understand how the input tensor is constructed; please explain more clearly):

We model each variant with four categorical variables:

- $A \in \{A, C, G, T\}$ is the alternate base at a SNP, or the reference base otherwise.
- $Z \in \{\text{homozygote, heterzygote}\}$ is the zygosity of the variant.
- $T \in \{\text{reference, SNP, insertion, delete}\}$ is the variant class.
- $L \in \{0, 1, 2, 3, 4, \geq 5\}$ is the length of an INDEL, where ' ≥ 5 ' represents a gap 5bp or longer.

For the truth data, each variable can be represented by a vector (i.e. 1-D tensor) using the one-hot or probability encoding, as is typically done in deep learning: $a_b = \Pr\{A = b\}$, $z_i = \delta(i, Z)$, $t_j = \delta(j, T)$ and $l_k = \delta(k, L)$, where $\delta(i, j)$ equals one if $i = j$, or 0 otherwise. The four vectors (a, z, t, l) are the outputs of the network.

The input of the network encodes allele counts. (reviewer comment: the authors said "the second tensor encodes the inserted sequences", but in authors' formulation, there is only one input tensor x . To describe a tensor, they need to explain what each dimension represents. I understand in $33 \times 4 \times 4$ '33' corresponds to position. I guess the first '4' corresponds to A/C/G/T on the reference (or on read?) and the second '4' to the four allele types as variable T above. However, I still don't understand how the relative counts are derived – what's the count at (A, insertion)? The authors should show an example about how these $4 \times 4 = 16$ numbers are calculated in exact. How to translate real data into tensors is one of the most important aspects in deep learning.)

With deep learning, we seek a function $F: x \rightarrow (a, z, t, l)$ that minimizes the cost:

$$C = \frac{1}{N} \sum_v \left(\sum_{b=1}^4 \left(\hat{a}_b^{(v)} - a_b^{(v)} \right)^2 - \sum_{i=1}^2 z_i^{(v)} \log \hat{z}_i^{(v)} - \sum_{j=1}^4 t_j^{(v)} \log \hat{t}_j^{(v)} - \sum_{k=1}^6 l_k^{(v)} \log \hat{l}_k^{(v)} \right)$$

where v iterates through all variants and a variable with a caret indicates it is an estimate from the network.

RE: We appreciate your patient guidance, thank you! We have updated the description accordingly. We have also elaborated how the input is generated in two places, 1) the "Clairvoyante" section in main text using layman terms, and 2) the "Pseudo code for generating the input" section in Supplementary Material using pseudo-code. We agreed that how to translate real data into tensors is one of the most import aspects in deep learning, we hope the two sections could accommodate both layman users and technical experts.

The following are minor comments:

6. I believe the method description also contains minor errors or subtleties. In no particular order:
- In authors' notation, a_i "denotes the probability of four possible reference and alternative alleles". How to set a_i given an INDEL and a multi-allelic SNP?
 - Does the model use base quality?
 - Cross-entropy takes the form of $\sum_i z_i \log \hat{z}_i$, where z_i is the truth. In authors' equation, z_i and \hat{z}_i are flipped.

RE: We have added text " a_b is set to all zero for an insertion or deletion. In the current Clairvoyante implementation, 1) multi-allelic SNPs are excluded from training, and 2) base-quality is not used (reasons and solutions in the "Discussion" section)" to elaborate the first and the second point. The third point has been corrected in the new description.

7. It would be good to evaluate using the syndip dataset (<https://github.com/lh3/CHM-eval>). GIAB is highly biased towards easy regions and is not a good representative of real applications.

RE: We have benchmarked Clairvoyante using the Syndip dataset both for model training and performance evaluation. Syndip is indeed an excellent truth variant dataset in addition to the GIAB datasets, especially when regarding the variants in the hard regions. Thus, in the future, we will include Syndip for creating pre-trained models so as to enhance the models' performance in the hard regions.

8. P2L64 (page 2, line 64). The initial deepVariant had to encode alignment as an image as it was using Google's DisBelief framework which only takes images as input. However, the latest deepVariant is built on top of TensorFlow as well. It doesn't have the old limitation any more.

RE: We have changed our wording and further elaborated the possibility of changing the input in the latest version of DeepVariant.

9. P9L327. It is not necessary to evaluate paf tools from minimap2. It was not designed to call variants from raw reads.

RE: We have removed paf tools from the evaluation.

10. P11L396. I am not sure how numbers are added up. The authors chose 100 FPs, of which 71 FPs are caused by multi-allelic sites. Then later the authors claimed only 12 FPs among the 100 FPs are true Clairvoyante errors. Wouldn't those 71 FPs be all Clairvoyante errors, too?

It is worth noting that variant callers are responsible for identifying alignment artifacts, which are often associated with distinct features (e.g. higher depth and higher variant density). Failing to detect alignment errors is still an error. GIAB is highly accurate. Based on the precision of Clairvoyante, I don't think 20% of differences could be GIAB errors. In addition, GIAB was constructed from a variety of data sources. Variants ambiguous in Illumina data may be clear in data sequenced with other technologies. How to 'validate' variants in IGV is also tricky. RTG's vcfEval may do sophisticated variant realignment, which may not be obvious to eyes. Overall, I think the discussion in this paragraph is too defensive. GIAB has done a phenomenal job on eliminating errors. If the authors want to investigate GIAB errors, they have to look at the results of multiple other variant callers, other short-read callsets and other technologies.

RE: We have rewritten the paragraph to fully recognize the errors made by Clairvoyante and to reflect on how the analysis of false positives and false negatives could help to improve Clairvoyante further.

11. P16L552. 3,135 missing variants in GIAB is an overestimate. There are at least two errors in the authors' analysis: GIAB contains variants at 13:104270904 and 10:65260789. Most other examples also look dubious. For example, on a different dataset, there are multiple 2bp deletions at 20:3,200,689 in Illumina data. 7:89312043 has multiple SNPs in a small region. This is a typical sign of false alignment. 1:566371 is located in the middle of repeats and could be caused by misalignment. X:143214235 shows extreme strand bias in Illumina data. I need to see the strands of long-read data and mapping quality to tell if it has a similar issue. The only convincing example where GIAB is wrong is 2:163811179. To this end, my observation strongly contradicts the authors' claim that "all ten variants ... are strongly supported by the data." Overall, the authors should weaken their claim in this paragraph. GIAB may miss variants, but there should be much fewer than 3,135.

RE: We have rewritten the paragraph to address these comments.

12. At last, the authors should give DeepVariants more credits. DeepVariants uses a much more sophisticated, though much slower, model and significantly outperforms GATK on the GIAB benchmark, which has been independently confirmed by multiple groups. DeepVariants also works for PacBio data. I know DeepVariants is not published yet, but I hope the authors may consider to evaluate it as well. A good paper should consider all available tools, peer-reviewed or not.

RE: We benchmarked DeepVariant on both the Illumina and ONT dataset in the revision, and include the latest citation to the now published algorithm. The results of the new benchmark are in the new section titled "Benchmarks of other state-of-the-art variant callers", which shows the high performance of DeepVariant on Illumina data along with the results of other variant callers including "Vardict", "LoFreq", and "FreeBayes" as requested by reviewer one.

Reviewers' comments:

Reviewer #1 (Remarks to the Author):

I really appreciate effort and time authors made to answer our comments. They made comparison with several other tools.

However I still have several doubts regarding this manuscript.

1. The authors state "(line 624)...On ONT data (rel5), DeepVariant performed much better than traditional variant callers that were not designed for long reads but it performed worse than Clairvoyante (Table 8)...". Yet, the results for Clairvoyante are not present in this table. For supporting the above claim it is necessary to prove it on the same test with the same data even if it is just a subset. I have not found the same performance benchmark for Clairvoyante in the manuscript (chr1 to chr22, chr22 to chr 1 and chr 21 to chr 22). Since Clairvoyante uses a simpler model than DeepVariant it is expected that it is faster.

2. The authors state "(line 398) ... As previously reported, DeepVariant has benchmarked the same dataset in their studied and reported 97.25% precision, 88.51% recall and 92.67% F1-score..." . This results are copied from the DeepVariant paper. However it is not clear whether the same training and test sets are used. For comparison the same datasets should be used. In addition there is a most recent version of DeepVariant. The original paper uses Version 0.4.1 and the current version is 0.7. The authors of Clairvoyante have already used version 0.6.1. I am aware that it is difficult to evaluate the most recent version in each revision so the version 0.6.1 should be a good enough. Also if base qualities are not provided in the datasets they could use dummy base qualities. It would decrease performances of other tools but we would have an idea which performances they could achieve.

3. Even if it is possible to prove that Clairvoyante performs better than deepVariant on long reads the authors should convince the readers that there is a need for a tool that would use PacBio or Nanopore reads and achieve much lower performances in comparison with the results deepVariant achieves on Illumina datasets.

Some minor comments.

1. Tables are unnecessarily detailed and because of that difficult to read. The authors could choose one model architecture, training parameters and candidate variant criteria for each technology. All other results could be in supplementary materials.

2. In the sentence "(line 99) ... containing 3,077,510 SNPs, 249,492 insertions and 256,137 deletions for GRCh37, 100 3,098,941 SNPs, 239,707 insertions and 257,019 deletions for GRCh37." GRCh37 is mentioned two times. I suppose one of these should be replaced with GRCh38.

3. In the sentence "(line 398) ...As previously reported, DeepVariant has benchmarked the same dataset in their studied and reported 97.25% precision, 88.51% recall and 92.67% F1-score..." the word studied should probably be replaced by study or studies.

Reviewer #2 (Remarks to the Author):

The authors now benchmarked trained model (without chr1) on chromosome 1, added note to why using UG, added nanopore's results on whole genome, added comparison to deep variant, separated

SNPs and Indels accuracies.

For table 8, in the ONT section, it has DeepVariant, Vardict and LoFreq, but is it missing Clairvoyante itself? it should add to the table for comparison if missing.

Reviewer #3 (Remarks to the Author):

The revised manuscript by Luo et al is much improved over the original version. The method description becomes cleaner, too. Nonetheless, I still have one major concern and a few minor comments.

1) I raised the question on "known sites" in the first of review (point 3). The authors explained their rationale without changing the results. I am afraid that their response is inadequate. In particular, the authors pointed out that Table 2 and 3 are meant to evaluate genotyping accuracy, but these two tables are not evaluating genotyping accuracy we care about. When we genotype at a set of SNPs, we don't know if the sample has the variant alleles before hand. We genotype the sample to find the answer. In the authors' experiment, they know all sites being typed are true variants. The two tables will mislead readers to believe they can achieve the reported accuracy. They can't.

What the authors should do is to take a list of common variants (e.g. >5% MAF in 1000g or from Illumina 550k genotyping chip) and report the genotyping accuracy on these sites. The authors may present a 3-by-3 table, where a row is indexed by ref/het/hom in the truth data and a column by ref/het/hom in the Clairvoyante calls. This table shows how often a true genotype is called differently by Clairvoyante. It evaluates genotyping accuracy. Alternatively, the authors may summarize the table with two numbers: the fraction of non-variant sites called as variants (i.e. FPR, false positive rate) and the fraction of variants missed (i.e. FNR, false negative rate). Don't report FDR (false discovery rate) or precision (=1-FDR) only. For genotyping, we want to know how often a non-variant site gets called as a variant. That is FPR, not FDR. I request the authors to show the genotyping accuracy, including FPR, on common variants and remove Table 2 and 3 (or move them to the supplementary). DeepVariant and LoFreq should also be included in the same table/figure.

Minor comments:

2) The bioconda command line in Clairvoyante's README doesn't work for me. I managed to manually install Clairvoyante on a machine supporting AVX2.

3) On syndip chr11, unfiltered GATK-HC gives me indel FNR=8.18% and FDR=5.47%. The corresponding metrics for Clairvoyante are 19.48% and 18.40%, respectively. This confirms the authors' observation that Clairvoyante has high error rate on syndip. This is a little worrying, though, given that GATK-HC can do much better.

4) Table 8 should show Clairvoyante and DeepVariant on Nanopore and PacBio data. Better merge Table 7 and 8 if possible.

Reviewer: Heng Li <hli@jimmy.harvard.edu>

To all reviewers

RE: We were able to decrease the running time of Nanopolish on whole genome from 40 days to 4.5 days using 24 CPU cores and a peak of 160GB memory. We have updated the tables and descriptions accordingly.

Reviewer: 1

I really appreciate effort and time authors made to answer our comments. They made comparison with several other tools.

However I still have several doubts regarding this manuscript.

1. The authors state "(line 624)...On ONT data (rel5), DeepVariant performed much better than traditional variant callers that were not designed for long reads but it performed worse than Clairvoyante (Table 8)...". Yet, the results for Clairvoyante are not present in this table. For supporting the above claim it is necessary to prove it on the same test with the same data even if it is just a subset. I have not found the same performance benchmark for Clairvoyante in the manuscript (chr1 to chr22, chr22 to chr 1 and chr 21 to chr 22). Since Clairvoyante uses a simpler model than DeepVariant it is expected that it is faster.

RE: We thank the reviewer again for their careful reading and further comments. We have added three benchmarks to the table. In the revision, we have also moved Table 8 to Supplementary Table 3 to improve the clarity of the presentation.

2. The authors state "(line 398) ... As previously reported, DeepVariant has benchmarked the same dataset in their studied and reported 97.25% precision, 88.51% recall and 92.67% F1-score..." . This results are copied from the DeepVariant paper. However it is not clear whether the same training and test sets are used. For comparison the same datasets should be used. In addition there is a most recent version of DeepVariant. The original paper uses Version 0.4.1 and the current version is 0.7. The authors of Clairvoyante have already used version 0.6.1. I am aware that it is difficult to evaluate the most recent version in each revision so the version 0.6.1 should be a good enough. Also if base qualities are not provided in the datasets they could use dummy base qualities. It would decrease performances of other tools but we would have an idea which performances they could achieve.

RE: We attempted to add dummy base qualities (QV=10) to the PacBio dataset and used the synthetic dataset for training a DeepVariant model using version v0.6.1. However, DeepVariant failed with exploded training losses and a non-converged model. As adding dummy base qualities is a bit of a hack and might not be a suggested action by DeepVariant, we consider unreliable to report this negative result on DeepVariant. As DeepVariant primarily targets short read (accurate base qualities are provided) while Clairvoyante targets long read (base qualities are either equivocal or missing), we would expect our current benchmarks to show the readers what they could expect from each tool as they originally designed.

3. Even if it is possible to prove that Clairvoyante performs better than deepVariant on long reads the authors should convince the readers that there is a need for a tool that would use PacBio or Nanopore reads and achieve much lower performances in comparison with the results deepVariant achieves on Illumina datasets.

RE: This is also a concern by reviewer 3. In order to clarify the performance to our readers, we have moved the previous "Call variants at known sites" benchmarks to the supplementary and added new benchmarks "Call variants at common variant sites" to demonstrate the expected performance of Clairvoyante on a typical precision medicine application where only clinically relevant or actionable variants are being genotyped. This application is becoming increasingly important in recent days as Single Molecule Sequencing is becoming more

widely used for clinical diagnosis of structural variations, but at the same time, doctors and researchers also want to know if there exist any actionable or incidental small variants without additional short read sequencing (Leija-Salazar et al., 2018). Clairvoyante's SNP calling F1-score on PacBio data reached 99.28% for HG001 and 99.30% for HG002, making Clairvoyante suitable for genotyping SNPs sensitively and accurately at known clinically relevant or actionable variants using PacBio reads in precision medicine application.

Some minor comments.

1. Tables are unnecessarily detailed and because of that difficult to read. The authors could choose one model architecture, training parameters and candidate variant criteria for each technology. All other results could be in supplementary materials.

RE: As also suggested by reviewer 3, we have moved Tables 5-8 to the supplementary materials.

2. In the sentence "(line 99) ... containing 3,077,510 SNPs, 249,492 insertions and 256,137 deletions for GRCh37, 100 3,098,941 SNPs, 239,707 insertions and 257,019 deletions for GRCh37." GRCh37 is mentioned two times. I suppose one of these should be replaced with GRCh38.

RE: We fixed the typos.

3. In the sentence "(line 398) ...As previously reported, DeepVariant has benchmarked the same dataset in their studied and reported 97.25% precision, 88.51% recall and 92.67% F1-score..." the word studied should probably be replaced by study or studies.

RE: We fixed the typos.

Reviewer: 2

The authors now benchmarked trained model (without chr1) on chromosome 1, added note to why using UG, added nanopolish's results on whole genome, added comparison to deep variant, separated SNPs and Indels accuracies.

For table 8, in the ONT section, it has DeepVariant, Vardict and LoFreq, but is it missing Clairvoyante itself? it should add to the table for comparison if missing.

RE: We have added the Clairvoyante results to the table. In the revision, we have moved Table 8 to Supplementary Table 3

Reviewer: 3 (Heng Li, hli@jimmy.harvard.edu)

The revised manuscript by Luo et al is much improved over the original version. The method description becomes cleaner, too. Nonetheless, I still have one major concern and a few minor comments.

1) I raised the question on "known sites" in the first of review (point 3). The authors explained their rationale without changing the results. I am afraid that their response is inadequate. In particular, the authors pointed out that Table 2 and 3 are meant to evaluate genotyping accuracy, but these two tables are not evaluating genotyping accuracy we care about. When we genotype at a set of SNPs, we don't know if the sample has the variant alleles before hand. We genotype the sample to find the answer. In the authors' experiment, they know all sites being typed are true variants. The two tables will mislead readers to believe they can achieve the reported accuracy. They can't.

What the authors should do is to take a list of common variants (e.g. >5% MAF in 1000g or from Illumina 550k genotyping chip) and report the genotyping accuracy on these sites. The authors may present a 3-by-3 table, where a row is indexed by ref/het/hom in the truth data and a column by ref/het/hom in the Clairvoyante calls. This table shows how often a true genotype is called differently by Clairvoyante. It evaluates genotyping accuracy. Alternatively, the authors may summarize the table with two numbers: the fraction of non-variant sites called as variants (i.e. FPR, false positive rate) and the fraction of variants missed (i.e. FNR, false negative rate). Don't report FDR (false discovery rate) or precision (=1-FDR) only.

For genotyping, we want to know how often a non-variant site gets called as a variant. That is FPR, not FDR. I request the authors to show the genotyping accuracy, including FPR, on common variants and remove Table 2 and 3 (or move them to the supplementary). DeepVariant and LoFreq should also be included in the same table/figure.

RE: After two rounds of email exchanges with the reviewer, we now fully understood and agreed with reviewer's comment. We have modified the manuscript accordingly. Using the 1kg phase 3 $MAF \geq 5\%$ variants as common variants and we benchmarked Clairvoyante, Nanopolish, DeepVariant and LoFreq. The new results are in a new section named "Call Variants at Common Variant Sites". We now report FPR and FNR in addition to other metrics. Old benchmarks and tables on known variants are removed. The unit tests were moved to supplementary materials.

Minor comments:

2) The bioconda command line in Clairvoyante's README doesn't work for me. I managed to manually install Clairvoyante on a machine supporting AVX2.

RE: We have added steps to the README to install from source code on machines without AVX2 support. We have been able to install Clairvoyante via bioconda on a few other systems so perhaps there are system-specific issues at play. If you forward your log files, we would be happy to assist.

3) On syndip chr11, unfiltered GATK-HC gives me indel FNR=8.18% and FDR=5.47%. The corresponding metrics for Clairvoyante are 19.48% and 18.40%, respectively. This confirms the authors' observation that Clairvoyante has high error rate on syndip. This is a little worrying, though, given that GATK-HC can do much better.

RE: As DeepVariant is doing great on short read, we will keep Clairvoyante's focus on optimizing for long reads. With additional resources in the future, we will drill deep into how subtle changes to the network parameter or architecture link to their performance of different inputs.

4) Table 8 should show Clairvoyante and DeepVariant on Nanopore and PacBio data. Better merge Table 7 and 8 if possible.

RE: This is also reviewer 1 and reviewer 2's concern. We copied the Clairvoyante results from Table 7 to Table 8 for a more straightforward comparison. We've also move Table 7 and Table 8 to supplementary materials.

REVIEWERS' COMMENTS:

Reviewer #1 (Remarks to the Author):

The authors answered comments. I think that in this form, the manuscript is correct and it presents some new ideas in using deep learning in variant calling. On ONT data, it performs better than Deep Variant, their major competitor. In addition, unlike Deep Variant, it does not need information about base quality for PacBio data. Therefore, I deem this manuscript deserves to be published.

However, I am still concerned whether Clairvoyante as a tool would be used for variant calling in practice. Illumina reads are cheap. Furthermore, in a recent paper Wenger et al have showed that Deep Variant performs quite well on long high quality Pacbio reads (Sequel System 6.0).

Reviewer: 1

The authors answered comments. I think that in this form, the manuscript is correct and it presents some new ideas in using deep learning in variant calling. On ONT data, it performs better than Deep Variant, their major competitor. In addition, unlike Deep Variant, it does not need information about base quality for PacBio data. Therefore, I deem this manuscript deserves to be published.

However, I am still concerned whether Clairvoyante as a tool would be used for variant calling in practice. Illumina reads are cheap. Furthermore, in a recent paper Wenger et al have showed that Deep Variant performs quite well on long high quality Pacbio reads (Sequel System 6.0).

RE: We thank the reviewer for their further comments. We further investigated the cost per Gb of other available ONT products for massive production. The details are available at <https://nanoporetech.com/products/comparison>. Compare to MinION that produced the data in our study, which costs \$50 at 10Gb, PromethION costs \$7 at 100Gb (i.e., \$0.7 at 10Gb), which is about 71 times lower than the MinION cost and comparable to Illumina's cost. PromethION has been shipped to a few laboratories around the world reports available online so far mostly confirm the massively reduce cost compared to its predecessor MinION. So, we believe Clairvoyante will have its position in Single Molecule Sequencing soon. We read Wenger et al. preprint published on Jan 13, 2019, in bioRxiv. Although we were unable to find the computational resource consumption of DeepVariant in the paper, we will make contact with the authors right away to build a model on PacBio's new data to provide the community with a much efficient solution than DeepVariant.